# Structural and electrophysiological basis for the modulation of KCNQ1 channel currents by ML277

Katrien Willegems[1,2,4], Jodene Eldstrom [1,4], Efthimios Kyriakis[1,4], Fariba Ataei[1,4], Harutyun Sahakyan [3], Ying Dou[1], Sophia Russo[1], Filip Van Petegem [2✉] & David Fedida [1✉]

The KCNQ1 ion channel plays critical physiological roles in electrical excitability and $K^+$ recycling in organs including the heart, brain, and gut. Loss of function is relatively common and can cause sudden arrhythmic death, sudden infant death, epilepsy and deafness. Here, we report cryogenic electron microscopic (cryo-EM) structures of *Xenopus* KCNQ1 bound to $Ca^{2+}$/Calmodulin, with and without the KCNQ1 channel activator, ML277. A single binding site for ML277 was identified, localized to a pocket lined by the S4-S5 linker, S5 and S6 helices of two separate subunits. Several pocket residues are not conserved in other KCNQ isoforms, explaining specificity. MD simulations and point mutations support this binding location for ML277 in open and closed channels and reveal that prevention of inactivation is an important component of the activator effect. Our work provides direction for therapeutic intervention targeting KCNQ1 loss of function pathologies including long QT interval syndrome and seizures.

[1] Department of Anesthesiology, Pharmacology and Therapeutics, University of British Columbia, Vancouver, BC, Canada. [2] Department of Biochemistry and Molecular Biology, University of British Columbia, Vancouver, BC, Canada. [3] National Center for Biotechnology Information, National Library of Medicine, National Institutes for Health, Bethesda, MD, USA. [4]These authors contributed equally: Katrien Willegems, Jodene Eldstrom, Efthimios Kyriakis, Fariba Ataei. ✉email: petegem@mail.ubc.ca; david.fedida@ubc.ca

KCNQ1 (Kv7.1) is the pore-forming subunit of potassium ion channel complexes that also comprise KCNE accessory subunits[1,2], calmodulin (CaM)[3,4], ATP, and phosphatidylinositol 4,5-bisphosphate (PIP2)[5,6]. Activators of KCNQ1 are important compounds as they may have pathophysiological and therapeutic importance given the consequences of KCNQ1/KCNE ion channel dysfunction, which include sudden cardiac death, seizures, loss of cochlear K$^+$ recycling, decreased gastrointestinal motility, type 2 diabetes, and renal dysfunction[7–9].

KCNQ1 has the structure of a canonical voltage-gated potassium (Kv) channel, consisting of four identical six-transmembrane helical domain subunits, arranged in a domain-swapped manner[4], and the significant functional diversity of the channel complex is largely accounted for by the presence of variable numbers of KCNE1-KCNE5 accessory subunits assembling as homomers[10] or heteromers[11]. For KCNE1 at least, the subunit ratios of KCNE1:KCNQ1 can be between 1:4 and 4:4, which dramatically affects the channel's pharmacology and current kinetics[10,12–14]. CaM co-purifies with the C-terminus of KCNQ1[15], it has complex effects on channel kinetics[16], and is a required accessory protein in the complex[3,4] where it is needed for trafficking to the plasmalemma[15,16], folding, and tetramerization[3]. ATP[17] and PIP2 are also required for channel activity, and binding of the lipid in the region of the S4-S5 linker[5] is largely responsible for coupling of voltage-sensing domain (VSD) activation to pore opening[6,18,19].

Activators of KCNQ1 channels exhibit a spectrum of activity that is often dependent on the number and type of accessory KCNE subunits in the channel complex[20]. ML277 ((2R)-N-[4-(4-methoxyphenyl)−2-thiazolyl]−1-[(4-methylphenyl)sulfonyl]−2-piperidinecarboxamide)[14,21], zinc pyrithione[22] and L-364,373[23] are most active on KCNQ1, with reduced activity when KCNE1 subunits are present[24], while phenylboronic acid[25], hexachlorophene, stilbenes such as DIDS (4,4′-diisothiocyano-2,2′-stilbene disulfonic acid) and SITS (4-acetamido-4′-isothiocyanatostilbene-2,2′-disulfonic acid), diclofenac acid derivatives such as mefenamic acid[13,26,27], and fatty acids and their analogues are more potent on saturated KCNQ1/KCNE1 complexes, although some increase currents from KCNQ1 alone as well[28–30]. ML277[21] is a particularly interesting and potent KCNQ1 activator, which increases peak currents dramatically in mammalian cells by enhancing voltage-dependent activation and eliminating inactivation[14,31–33]. Mechanistically, these effects are considered to result from enhanced voltage sensor-pore coupling[33] that prolongs channel residency in the open state and reduces inactivation[32]. As mentioned above, the action of ML277 is dependent on the stoichiometry of the KCNQ1/KCNE1 complex, with studies finding little[31] or no effect[14,33] of the drug when KCNQ1 is saturated with KCNE1 (4:4 stoichiometry). As ML277 enhances cardiac delayed rectifier K$^+$ current (IKs) in human induced pluripotent stem cell-derived cardiomyocytes from normal individuals and from patients with long QT-interval syndrome (LQTS1)[14,34], the potential therapeutic relevance of ML277 is clear, as is the idea of unsaturated KCNE1:KCNQ1 stoichiometries in humans, perhaps 2:4, as has also been suggested from myocyte experiments in other mammals[31,35].

While we know of no structures at present of drugs in complex with KCNQ1, recent reports show structures of the neuronal KCNQ channels, KCNQ2[36] in complex with two activators, retigabine and ztz240, and KCNQ4 in complex with retigabine[37] and ML213[38]. These studies in KCNQ4 of smaller molecules than ML277 demonstrate the presence of elongated densities with overlapping binding sites oriented axially alongside the S5 and S6 helices, in pockets defined by the S5, S6, and pore helices of more than one subunit protomer.

Given the potential importance and relevance of ML277 to KCNQ1/KCNE1 and its wider implications for modulation of IKs, here we show structures of *Xenopus* KCNQ1 with and without bound ML277, using cryo-EM. We observe that the structure of xKCNQ1 alone shows almost complete identity with the density determined by Sun and MacKinnon[4], highlighting the accuracy and reproducibility of the techniques used. The high degree of similarity between our apo structure and that of Sun and MacKinnon, suggests that an extra density we see at the proposed binding site when cryo-EM samples are incubated in ML277 is due to the presence of the drug. We find that ML277 resides in a pocket defined by the S4-S5 linker helix, the transmembrane (TM)-spanning S5 and S6 helices from one protomer and the S5′ and S6′ helices from an adjacent protomer. A twist observed at the carboxy-terminal domain is also likely the result of a conformational change due to ML277. To probe the action of specific residues involved in the binding of KCNQ1 to ML277, we mutate individual residues of interest using site-directed mutagenesis. The electrophysiological data support the described binding site for ML277 and also support the idea that removal of inactivation is a major mechanism through which ML277 exerts its action. Mutations of residues that we find involved in the specific binding of ML277 to KCNQ1 have been implicated in cases of Jervell and Lange-Nielsen syndrome, an autosomal dominant form of LQTS1[39], as well as in action potential prolongation in patient-derived cardiomyocyte iPSCs[40]. Thus, the present findings have important implications for understanding functional mechanisms of activation and inactivation of KCNQ1 channels, the underlying effects of disease-causing mutations in patients, and also provide insight into potential sites for activator action that can have therapeutic relevance.

## Results

**Structures of xKCNQ1-CaM in the presence and absence of ML277.** We utilized a biochemically stable form of *Xenopus* KCNQ1 (xKCNQ1, residues 67–610), bound to Ca$^{2+}$/CaM[4] to perform cryo-EM studies in the absence and presence of ML277 (xKCNQ1-CaM and xKCNQ1-CaM-ML277, respectively)[21]. The functional response of the truncated construct to ML277 was tested and showed typical human full-length KCNQ1 characteristics (Supplementary Fig. 1). Note that residue numbering is +10 in humans compared with xKCNQ1, and to distinguish them we use a single letter code to denote residues in hKCNQ1 rather than the three-letter code we use for xKCNQ1.

Control xKCNQ1 currents activated rapidly and showed a small subsequent inflection that was indicative of ongoing inactivation (Supplementary Fig. 1a). The half-activation voltage ($V_{1/2}$) was −10 mV, similar to that reported for hKCNQ1 expressed in mammalian cells (Supplementary Fig. 1b, Supplementary Fig. 9, Supplementary Table 3), and tail currents during repolarization showed a typical hook upon recovery from inactivation (Supplementary Fig. 1c). Truncated xKCNQ1 also responded to 1 μM ML277 in a similar manner to hKCNQ1[14,32] with a large increase in peak and tail currents, a slowing of tail current decay, and a hyperpolarizing shift of the conductance-voltage (G-V) relationship (Supplementary Fig. 1a, b). Currents were highly sensitive to the specific KCNQ1 antagonist, HMR1556, with 1 μM capable of abolishing tail currents completely (or 5 μM in the presence of ML277 as well, Supplementary Fig. 1d, e).

The truncated xKCNQ1 was tagged with GFP and over-expressed together with CaM in tsA201 cells and purified using an anti-GFP nanobody-based affinity purification, with the final sample containing 0.5 mM CaCl$_2$. In xKCNQ1-CaM-ML277 samples, ML277 was included at 100 μM to ensure full saturation

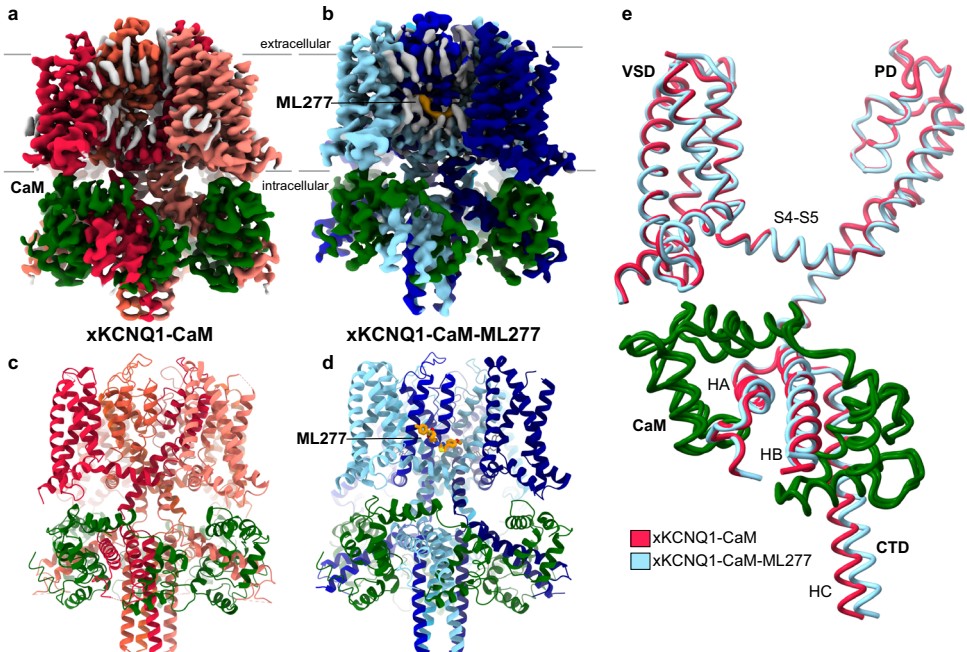

**Fig. 1 Cryo-EM maps of Xenopus KCNQ1-CaM alone and in complex with ML277. a** Cryo-EM density map of *Xenopus* KCNQ1-CaM (xKCNQ1-CaM), and its model structure (**c**). CaM is colored dark green and KCNQ1 subunits are in shades of red. **b** xKCNQ1-CaM in complex with ML277 and its model structure (**d**). CaM and ML277 are colored dark green and orange, respectively. KCNQ1 subunits are in shades of blue. Additional densities, likely representing unmodeled detergent or lipid molecules are shown in gray. For further analysis of these densities, see Supplementary Fig. 7. **e** Overall superposition of a single subunit of xKCNQ1-CaM and xKCNQ1-CaM-ML277 shows a visible twist in the C-helix coiled-coil domain (CTD). HA-HC refers to helices within the C-terminal domain of KCNQ1.

of its potential binding site(s). We obtained cryo-EM structures for xKCNQ1-CaM at 3.8 Å and for xKCNQ1-CaM-ML277 at 3.9 Å, both with C4 symmetry imposed (Fig. 1a–d, Supplementary Figs. 2–5). The transmembrane (TM) region was best resolved, reaching a local resolution of ~3 Å in both maps (Supplementary Figs. 2–3). The C-terminal coiled-coil displayed a resolution worse than 4 Å and was modeled until residue 556. The missing density for the remainder of the structure is likely due to the inherent flexibility of this region, an observation also made in other KCNQ structures[4,5]. The xKCNQ1-CaM structure is in the same decoupled configuration as the published xKCNQ1-CaM structure[4], with the voltage sensor domains in the activated state and the pore closed (Supplementary Fig. 3c). The xKCNQ1-CaM-ML277 structure (Fig. 1b, d) shows a similar decoupled TM configuration but a differently twisted C-terminal coiled-coil domain (Fig. 1e). In both structures, CaM was modeled with $Ca^{2+}$ ions bound to EF hands 1, 2, and 4.

**The binding pocket of ML277 and conformational changes**. The map for xKCNQ1-CaM-ML277 shows clear additional density in the TM region when compared to both our own xKCNQ1-CaM and a previously published xKCNQ1-CaM structure[4] (Figs. 1, 2, Supplementary Fig. 7). To verify this density, we also collected an additional, independent data set for xKCNQ1-CaM-ML277 (data set 2) from a separate protein sample (Supplementary Fig. 4), which showed a similar density in the same location (Supplementary Fig. 7d). Superposition of the unmodeled densities onto the xKCNQ1-CaM-ML277 structure revealed no other unique density in the xKCNQ1-CaM-ML277 map that could fit the elongated shape of ML277 (Supplementary Fig. 7e).

Thus, two independent data sets with ML277 show density for a ligand at the same location, whereas two data sets without

ML277 show the absence of density, which strongly supports the proposed location for ML277. The structure of ML277 is elongated, which matches the elongated shape of the density, and the orientation was chosen based on the highest map-to-model correlation coefficient[41] (Fig. 2a, c), although an alternative binding conformation cannot be ruled out completely. ML277 binds in a hydrophobic pocket of the TM region formed by the S4-S5 linker helix, the TM spanning S5 and S6 helices from one protomer and the S5′ and S6′ helices from an adjacent protomer (Fig. 2d). The ML277 binding pocket is lined by residues Trp238, Leu241, and Val245 in the S4-S5 linker; Leu252, Thr255, and Leu256 in helix S5; Pro333 and Leu337 in helix S6 from one protomer and residues Ile258′, Leu261′, Gly262′ and Phe265′ in helix S5′ and Phe322′, Val324′, Phe325′, Ala326′, Ser328′ and Phe329′ in helix S6′ from the neighboring protomer (Fig. 2d). Supplementary Table 1 shows the binding pocket in *Xenopus* KCNQ1 numbering vs human KCNQ1, and the other KCNQ isoforms.

The binding of ML277 does not trigger major structural changes in comparison to our own and a previously published unbound apo structure[4], with the pore remaining closed (Supplementary Fig. 3c). However, some conformational changes are observed, including a small displacement of the C-terminal coiled-coil domain (CTD) (Figs. 1e, Fig. 3). 3D classification and non-uniform refinement of one class and two pooled similar classes (at 4.1 and 4.8 Å resolution respectively) showed two different degrees of twisting of the coiled-coil domain, suggesting that the modeled conformational change in this region is a representation of a flexible turning of the coiled-coil. The twist of the CTD was also observed in the second independent data set that was collected in the presence of ML277 (Fig. 3). There are also local changes in residues that line the binding site. This includes a significant conformational displacement of the side chain of Leu256 to allow ML277 to bind, which would otherwise

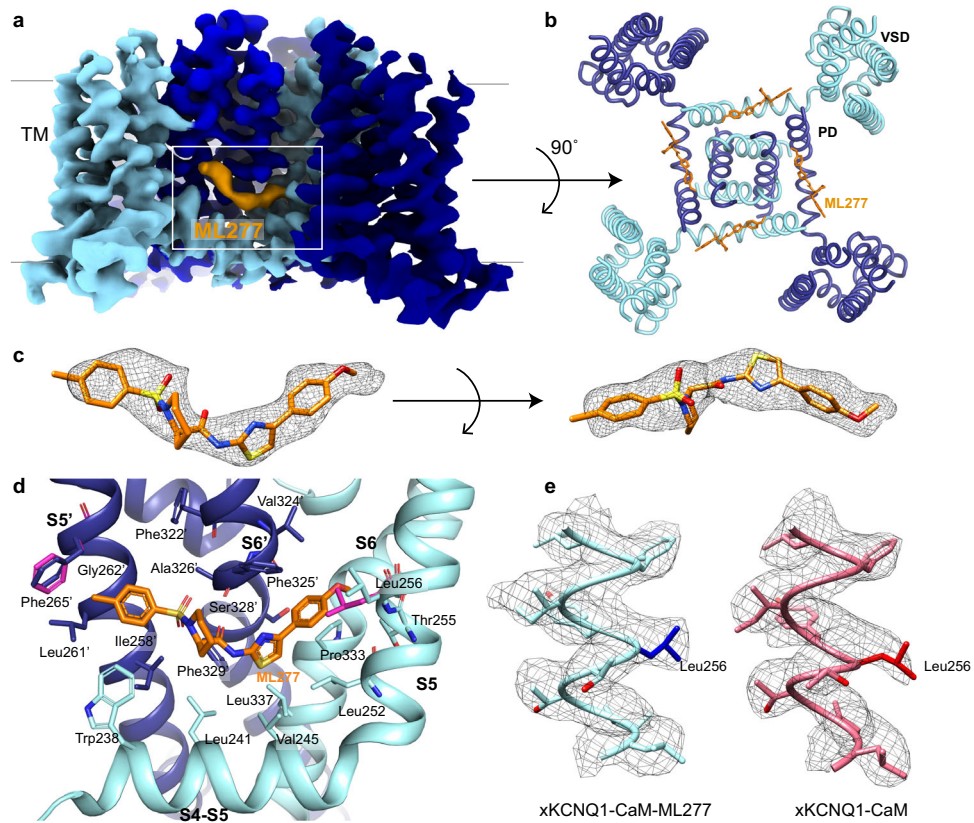

**Fig. 2 ML277 binding to xKCNQ1-CaM. a** The cryo-EM density map for xKCNQ1-CaM-ML277 with the density for ML277 indicated in orange. The density map is cut off at the TM region for clarity. **b** Top view of xKCNQ1-CaM-ML277 from the extracellular side is shown with ML277 (orange) bound in its hydrophobic pocket. The structure is clipped in the frontal plane at the level of the binding site for clarity. **c** Isolated density (5x rmsd) for ML277 is shown as a mesh with the ML277 docked inside in the orientation with the highest map-to-model correlation coefficient. **d** Enlarged side view of the ML277 binding pocket formed by the S4-S5 linker helix, the transmembrane spanning S5 and S6 helices from one protomer (light blue) and the S5′ and S6′ from an adjacent protomer (dark blue). The position of the side chain of Leu256 and the possible rotation of the phenyl ring of Phe265′ in the xKCNQ1-CaM structure are shown in pink. **e** Comparison of the density map (contour level at 5x rmsd) for xKCNQ1-CaM and xKCNQ1-CaM-ML277 focused on the S5 helix.

be hindered sterically (Fig. 2d, e). There is also a possible rotation of the phenyl ring of Phe265′, which was modeled in a position to allow space for the 4-methylphenyl group of ML277 (Fig. 2d). A movie summarizes the changes in xKCNQ1-CaM, especially the twist in the CTD, during the binding and unbinding of ML277 (Supplementary Movie 1).

**ML277 binding to the open channel in the presence of PIP2 and KCNE3.** As described above, the presence of ML277 in the xKCNQ1-CaM-ML277 structure mainly reveals small local changes in the binding pocket compared to the unbound structure. In contrast with this, the binding of PIP2 leads to substantial conformational changes in the open channel, including a rearrangement of the S6$_{helix}$–loop–HA$_{helix}$ segment to a continuous long helix with concomitant rotation of CaM by ~180°, which abolishes its interactions with the VSD[5]. Superposition of the structures of xKCNQ1-CaM-ML277 and hKCNQ1-CaM-PIP2 (PDB-ID: 6V01)[5], based on either the S4-S5 linker or the S6′ helix, reveals that the lipid tail of PIP2 potentially clashes with the 4-methylphenyl group of ML277 (Fig. 4a, c). ML277 may cause a small displacement of the PIP2 tails which are not as critical to activity as the headgroup placement[42]. This flexibility allows simultaneous binding of both molecules, as PIP2 is mandatory and irreplaceable for KCNQ1 pore opening[6,19]. The data agree with the model prediction and experiments of Xu et al.[31]

who suggested overlapping binding sites for ML277 and PIP2. There are also small shifts of residues directly within the ML277 binding site, ~1.4 Å for Leu241, ~0.8 Å for Pro333, ~0.7 Å for Gly262′, and ~1.0 Å for Phe325′. In addition, the side chains of Leu256 and Phe260 in the xKCNQ1-CaM-ML277 structure have different conformations in comparison to L266 and F270 in the hKCNQ1-CaM-KCNE3-PIP2 structure as a result of the binding of KCNE3 (Fig. 4a, c).

Docking of ML277 into the open pore hKCNQ1-CaM-KCNE3-PIP2 structure, with KCNE3 removed, and energy minimization of the protein-ligand complex, indeed suggests a displacement of the lipid tail to accommodate the methylphenyl group of ML277, but little change in the position of the PIP2 head group (Fig. 4b, d). The MD simulation and binding free energy calculations (Methods) revealed that ML277 can bind to the open channel state with a similar configuration as in the xKCNQ1-CaM-ML277 structure, with ΔG values of −47.6 ± 2.5 (closed) kcal/mol and −54.5 ± 2.9 kcal/mol (open; Supplementary Fig. 10). In addition, the lipid tail of PIP2 is very flexible and can adopt many different conformations to allow the binding of ML277. Simulations in Supplementary Fig. 8 show lipid tail poses adopted during MD simulation, illustrating the range of potential conformations of PIP2 tails and ML277 in hKCNQ1-CaM-KCNE3-PIP2 (PDB-ID: 6V01), with KCNE3 removed.

During MD simulations, the side chains of hKCNQ1 residues L266 and F270 (Fig. 4b) adopt the same conformation as in the

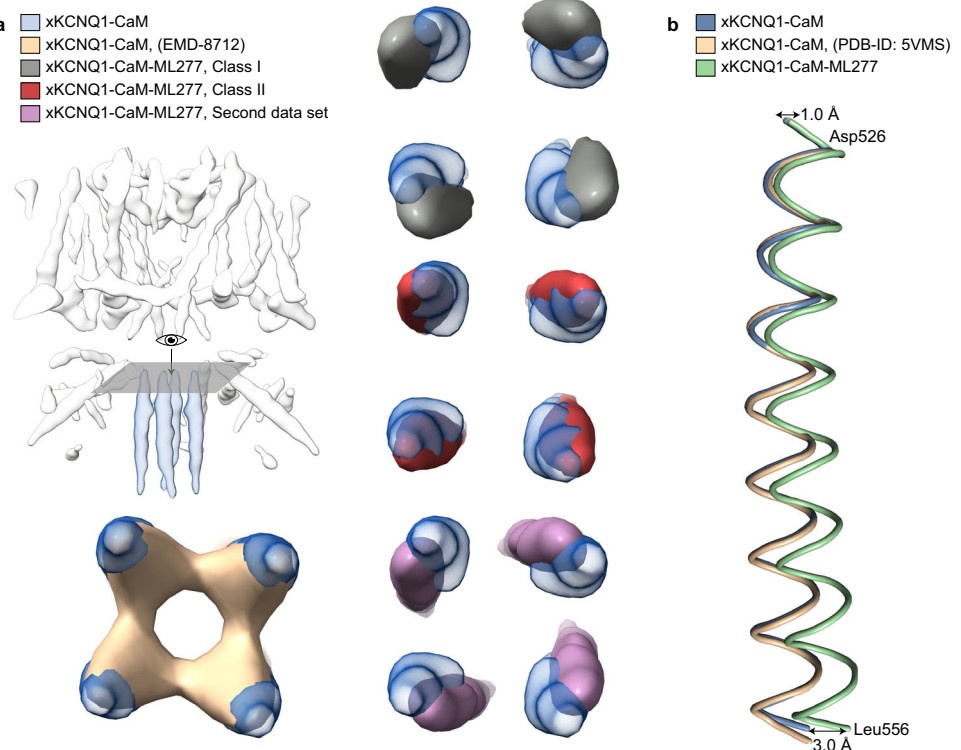

**Fig. 3 xKCNQ1-CaM-ML277 shows a twisted coiled-coil C-terminal domain. a** Heterogeneous refinement ($K = 4$) of xKCNQ1-CaM-ML277 data set 1 showed three KCNQ1 classes (see Supplementary Fig. 3). Two classes with similar twist were pooled and refined shown here as class I whereas the third class is shown as class II for clarity. Both classes have different CTD twist degrees between them, and compared to our native and previously studied xKCNQ1-CaM map (PDB-ID: 5VMS), indicating a flexibility in the ML277-induced twist of the coiled-coil CTD. The native structure which was determined in this work is shown in light blue. This twist was also observed for the xKCNQ1-CaM-ML277 data set 2. **b** The twist of the CTD in the ML277-bound xKCNQ1-CaM (light green) compared to our and previously studied native structures (blue and brown, respectively) shows a displacement of the backbone by ~1 Å at the top of the CTD (Asp526) and due to the propagated twist up to ~3 Å around the last modeled residue, Leu556.

experimentally-determined xKCNQ1-CaM-ML277 structure (Figs. 4a, 2d) highlighting the important role of Leu256 for ML277 binding in xKCNQ1, and of both L266/F270 in hKCNQ1 on the reduced efficacy of ML277 in the presence of KCNE3.

KCNQ1 pairs with KCNE3 to produce a constitutively active channel[43], and in the presence of saturating amounts of KCNE3, ML277 no longer activates the channel complex[14]. Structural comparison of the complex of xKCNQ1-CaM-ML277 and hKCNQ1-CaM-PIP2-KCNE3 (PDB-ID: 6V01)[5] suggests that the binding of KCNE3 and ML277 may be mutually exclusive (Fig. 4e, f). The binding of ML277 triggers a displacement of the side chain of Leu256 which is prohibited when KCNE3 is present. More specifically, the superposition of the complex of xKCNQ1-CaM-ML277 onto hKCNQ1-CaM-PIP2-KCNE3 (PDB-ID: 6V01) shows that the conformation of Leu256 in the xKCNQ1-CaM-ML277 structure would clash with residue T71 in hKCNE3 and F270 in hKCNQ1 (Fig. 4f). In the xKCNQ1-CaM-ML277 structure the chi1 angle (rotatable bond between Cα and Cβ) of Leu256 is 173°, and in the hKCNQ1-CaM-PIP2-KCNE3 structure is −72°. In the presence of KCNE3, if the chi1 angle is more negative than −80° it clashes with F270, and if it is higher than 50° it clashes with T71, which demonstrates that the chi1 angle in the presence of KCNE3 cannot be 173° as in the case of xKCNQ1-CaM-ML277.

Moreover, a potential displacement of the side chain of F270, in order to adopt a conformation similar to the xKCNQ1-CaM-ML277 structure, is unlikely to happen due to a potential clash with residue F68 in hKCNE3. This potential steric hindrance is further supported by the fact that the conformation of F270 in the hKCNQ1-CaM structure is similar to the xKCNQ1-CaM-ML277

structure and changes when hKCNE3 binds. The spatially restricted F270, as a result of KCNE3 binding, and the close proximity of KCNE3 residue T71[5], does not allow rotation of L266 and therefore would prevent ML277 from binding. Thus, KCNE3 seems to restrict access to ML277 by preventing the required conformational changes in its binding pocket.

**Determinants of KCNQ isoform and activator specificity.** All residues that line the binding pocket for ML277 are conserved between xKCNQ1 and human KCNQ1, but several residues are not conserved in other KCNQ isoforms, which likely determine the specificity of ML277 for KCNQ1 (Fig. 5). A full list of these differences is shown in Supplementary Table 1, but three of those seem particularly important.

The overall shape of the binding pocket of ML277 in KCNQ1 differs in hKCNQ2 and hKCNQ4 as shown in the surface representations (Fig. 5a–d). Leu256, which undergoes a rotation upon binding ML277, interacts with the 4-methoxyphenyl group of ML277 and is replaced by a bulkier Trp in all other KCNQ isoforms (Supplementary Table 1), W236 and W242 in KCNQ2 and KCNQ4, respectively (Fig. 5c, d), where it may cause steric hindrance to ML277 binding. The xKCNQ1 residue Gly262′, which is located in transmembrane helix S5′ (Fig. 2d) immediately adjacent to the 4-methylphenyl ring of ML277, is also replaced by bulkier residues like Cys, Thr, or Val in other KCNQ isoforms[36–38] which are not affected by ML277. Finally, Phe325′, which interacts with the internal piperidine ring system of ML277, is replaced by a smaller Leu or Ile in all other isoforms.

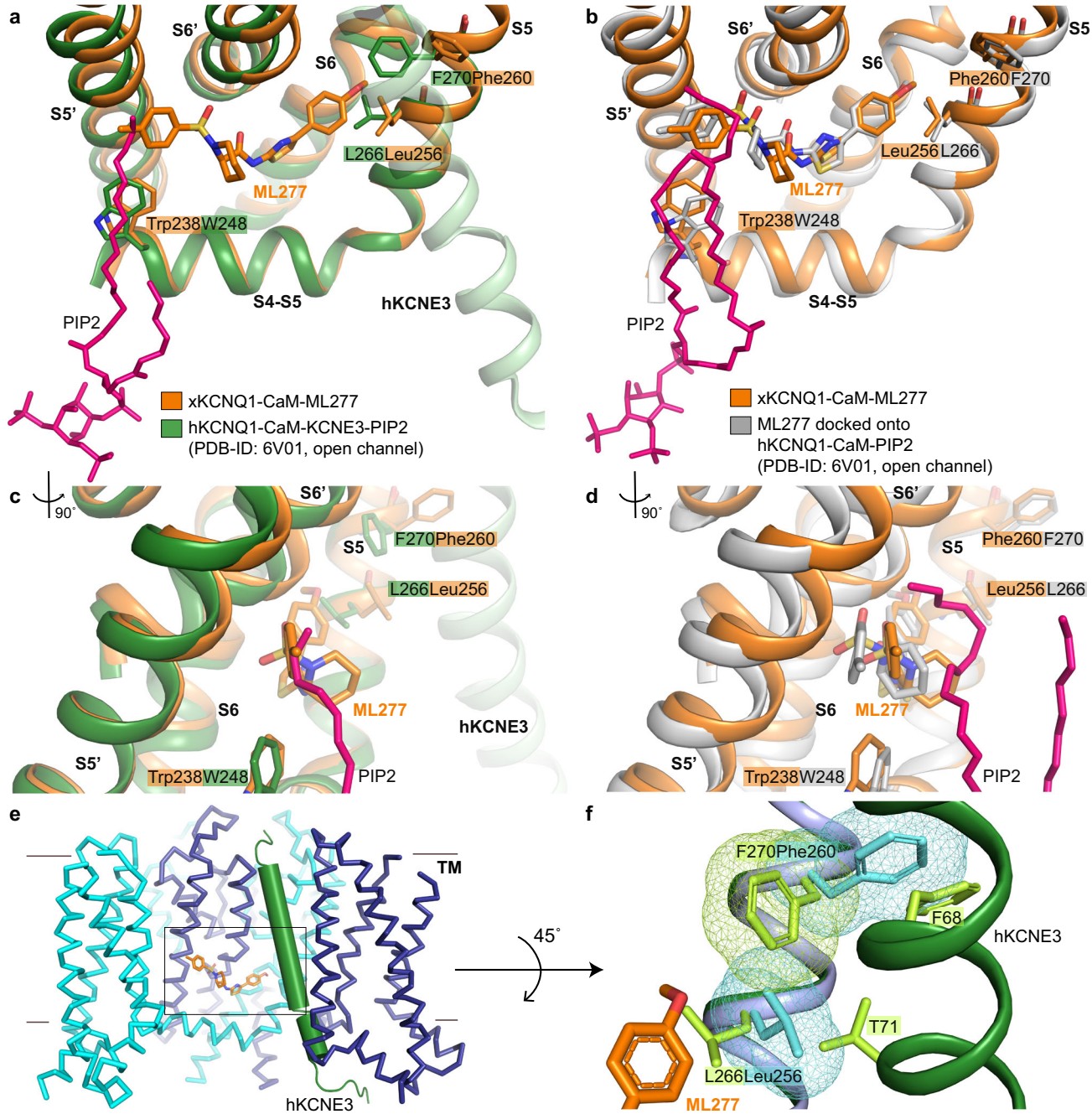

**Fig. 4 ML277 binding to the open hKCNQ1 structure in the presence of PIP2 and KCNE3. a, c** Superposition of xKCNQ1-CaM-ML277 structure (orange) onto the open pore hKCNQ1-CaM-KCNE3-PIP2 (PDB-ID: 6V01, green) via the S4-S5 linker and S5 helix. hKCNE3 is shown as pale green for comparison. **c** Side view of PIP2 tail and ML277. **b, d** Superposition of xKCNQ1-CaM-ML277 structure (orange) onto the modeled hKCNQ1-PIP2-ML277 structure (gray, PDB-ID: 6V01, KCNE3 was omitted). Binding free energy for ML277 was −54.5 ± 0.65 kcal/mol. **d** Side view of PIP2 tails and ML277. **e** Superposition of KCNE3 (green) from hKCNQ1-CaM-PIP2-KCNE3 (PDB-ID: 6V01), onto the xKCNQ1-CaM-ML277 complex (cyan and blue). **f** Conformations of T71 and F68 in KCNE3 and F270 in hKCNQ1 do not allow L266 to rotate and ML277 to bind. Residues on KCNQ1-CaM-PIP2-KCNE3 (green) are lime green and corresponding residues Leu256 and Phe260 in xKCNQ1-CaM-ML277 are shown in cyan. VDW surfaces of Leu256, Phe260 in xKCNQ1, and F270 in hKCNQ1 are shown as mesh to indicate potential clashes.

To verify that ML277 is not able to bind to KCNQ2 and KCNQ4, we performed molecular docking studies with side-chain flexibility (Fig. 5e, f). The Internal Coordinate Mechanics (ICM software) docking scores[44] for KCNQ2 and KCNQ4 were −16.6 and −9.42, respectively, which is much less favorable than for xKCNQ1-ML277 (−31.1). In addition, no docking poses for KCNQ2 and KCNQ4 were similar to that seen in the xKCNQ1-CaM-ML277 cryo-EM structure.

The importance of xKCNQ1 residues Leu256, Gly262′, and Phe325′ for ML277 selectivity is further supported by aligning the published locations of other KCNQ regulators with the xKCNQ1-CaM-ML277 structure (Fig. 6a, b). ML213 on KCNQ4 and retigabine (RTG) on KCNQ4 and KCNQ2 bind in the same pocket which partially overlaps with the ML277 binding site, whereas ztz240 binds to KCNQ2 in a separate voltage sensor location. W242 in S5 was previously shown to be important for

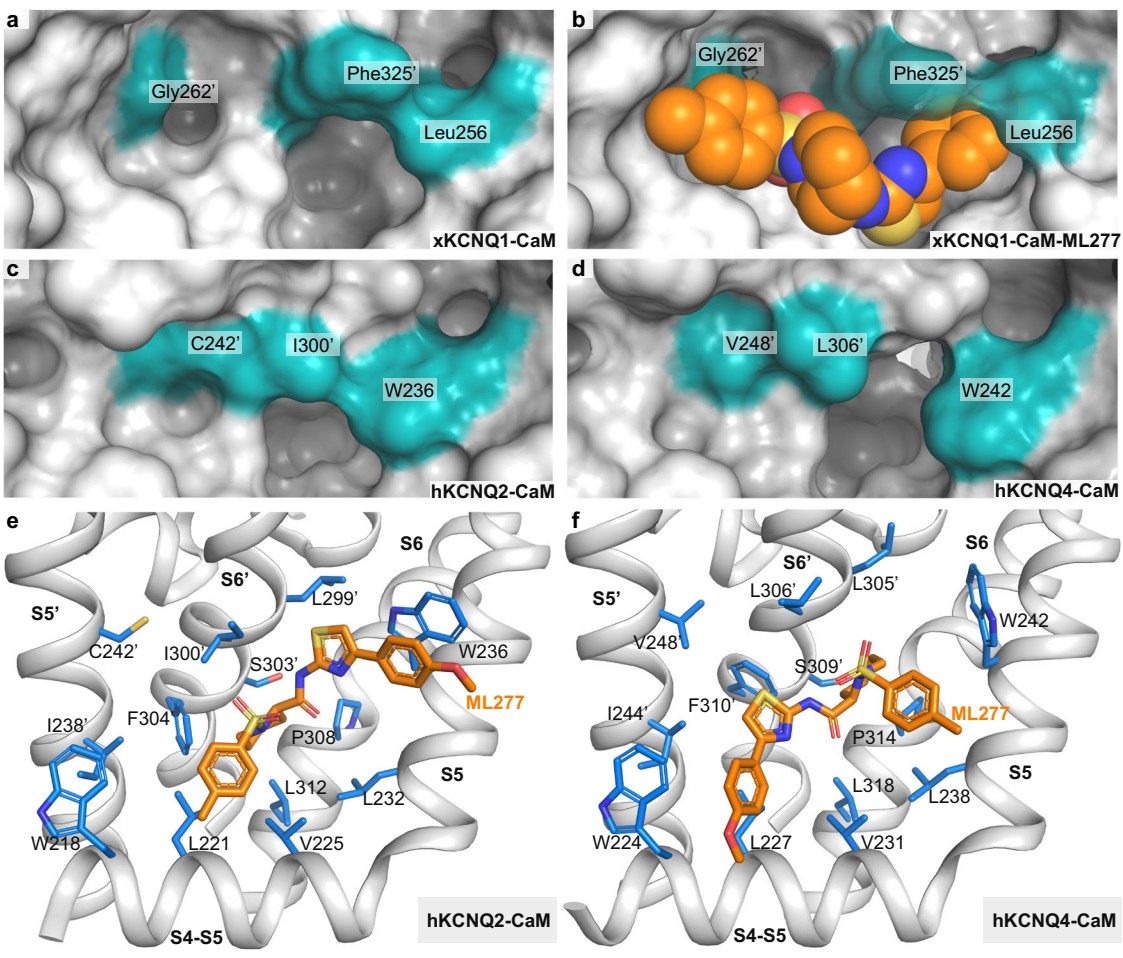

**Fig. 5 Binding of ML277 to xKCNQ1, hKCNQ2, and hKCNQ4. a–d** Surface representation of the ML277 binding pocket in xKCNQ1 (**a**, **b**), hKCNQ2 (PDB-ID: 7CR3) and hKCNQ4 (PDB-ID: 7BYL). Docking score for ML277 to xKCNQ1 was −31.1. **e** MD docking of ML277 onto KCNQ2 (7CR3). ML277 is shown in orange. Docking score was −16.6. **f** MD docking of ML277 onto hKCNQ4 (7BYL). Docking score was −9.42.

the binding of ML213[37,38] as well as for the binding of retigabine (RTG)[36,37]. Thus, individual substitutions may increase binding for ML213 and RTG, while reducing affinity for ML277, and vice versa, suggesting that the specificities of these small molecules are affected by the residues lining the binding pocket (Fig. 6c, d). Superposition of hKCNQ4-CaM-ML213 (PDB-ID: 7VNQ) onto xKCNQ1-CaM-ML277 shows that ML213 may clash with xKCNQ1 residues Leu256 (W242 in hKCNQ4), and Phe325′ (L306′ in hKCNQ4). Phe325′ in xKCNQ1 is replaced by a smaller Leu or Ile in all other isoforms (Supplementary Table 1). The superposition of xKCNQ1-CaM-ML277 onto hKCNQ4-CaM-ML213 shows a possible clash of ML277 with this Leu (Leu306′ in hKCNQ4), as well as with residues W242 and V248′ (Fig. 6d).

Taken together, our results highlight the importance of Leu256, Phe325′, and Gly262′ for both the isoform and activator selectivity of ML277 for KCNQ1.

**Functional effects of mutations to pocket residues validate the ML277 binding site.** To verify our identification of the binding pocket for ML277 in KCNQ1, we undertook a functional analysis of binding site mutants on human KCNQ1, which shows a very similar response to ML277 as truncated xKCNQ1 (Supplementary Fig. 1).

Three electrophysiological characteristics have been used by ourselves and others to quantify the action of ML277 on KCNQ1 channels and are illustrated by data in Fig. 7; these are, changes in

current amplitude, hyperpolarizing shifts in the half-activation potential, and a reduction in channel inactivation[14,31,32]. In the presence of ML277, wild-type (WT) currents increased during the step depolarization to +60 mV and tail currents increased ~8-fold on exposure to ML277, (Fig. 7a, c, Supplementary Fig. 9, Supplementary Table 2), which was also true of the V255A mutant (Fig. 7c). S338A was the only mutant in which the tail current increase in the presence of ML277 exceeded WT channels with a >16-fold augmentation of the tail current. L251A, L262A, L266A, L266W, G272A, and F335A tail currents increased from 1.6 to 3.6-fold in the presence of ML277, but this was significantly less than for WT and V255A channels (Fig. 7c, Supplementary Table 2). F339A showed no change with ML277 and S338F, G272C, G272T, G272L, and G272V peak and tail currents actually decreased in the presence of ML277 which suggests that in the absence of its facilitatory action, ML277 binding may inhibit KCNQ1 currents (Fig. 7c, Supplementary Fig. 9c, d).

ML277 also slows deactivation at −40 mV (Fig. 7a, black arrows)[14,31,32], and at −120 mV (Fig. 7b), which reflects changes in the kinetics of VSD movement at these potentials. To investigate this more broadly, the half-maximal activation voltage ($V_{1/2}$) was measured for WT and each of the binding site mutants before and after exposure to ML277. At an interpulse interval of 20 s the WT $V_{1/2}$ of activation was −35.3 ± 4.4 mV in the presence of 1 μM ML277, as indicated by the red point and line in Fig. 7d. L262A, L266A, L272A, and G272L each showed a similar ~15 mV negative shift in $V_{1/2}$ compared with control before

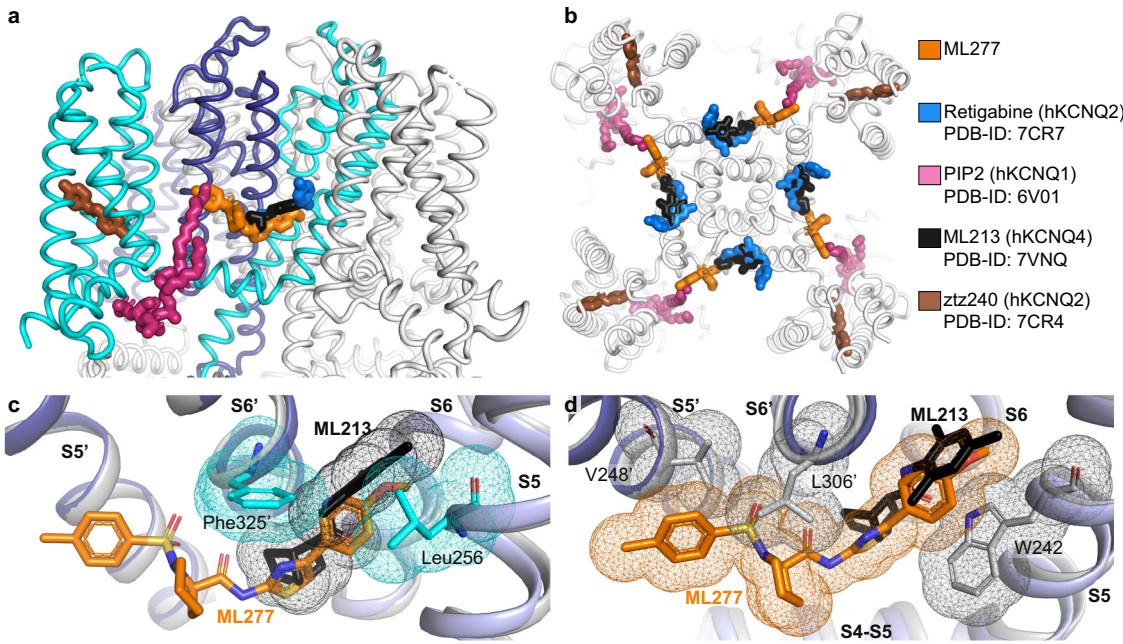

**Fig. 6 Binding of ML277 and other KCNQ regulators to xKCNQ1 and hkCNQ4. a, b** Overview of published KCNQ-ligand binding plotted on the xKCNQ1-CaM-ML277 structure. Different isoform structures were aligned based on the pore-forming domain. **a** View from within the plane of the membrane. Cyan and blue for the two subunits comprising the ML277 binding pocket. **b** Top view from the extracellular side clipped at the level of the ML277/ML213/retigabine binding sites for clarity. **c** Superposition of hKCNQ4-CaM-ML213 (gray) onto xKCNQ1-CaM-ML277 (different shades of blue for different subunits, xKCNQ1 residue side chains shown in cyan) ML277 is shown in orange, and ML213 in black. Van der Waals surfaces indicate potential clashes. **d** Superposition of xKCNQ1-CaM-ML277 onto hKCNQ4-CaM-ML213 (ML213 in black and hKCNQ4 residue side chains in gray). Van der Waals surfaces indicate potential clashes between ML277 and hKCNQ4 residues W242, L306′ and V248′.

ML277 (Supplementary Table 3). L251A, V255A, G272C, G272T, and G272V showed no shift and L266W, F335A, and F339A all showed depolarizing shifts in $V_{1/2}$ after ML277 treatment. Interestingly, S338A, which showed the largest current increase in the presence of ML277, showed only a small depolarization of the $V_{1/2}$ in ML277 (Fig. 7d, Supplementary Fig. 9, Supplementary Table 3).

It has been previously shown that ML277 prevents inactivation in KCNQ1 and that this is one of the mechanisms by which the drug increases current amplitude[14]. Inactivation is largely hidden during depolarizing steps in KCNQ1 and only becomes evident as a hook in the tail currents. WT tail currents show a prominent hook on repolarization at −120 mV that reflects channels recovering from inactivation before they deactivate (Fig. 7b, left panel). Tail current morphology was altered in mutant channels in the absence of ML277 such that the hook was lessened (F339A), or abolished in some (G272C in Fig. 7b right panel, L251A and G272T in Fig. 8a inset, G272T/L/V in Supplementary Fig. 9e). In S338A it can be seen that the tail current hook is slower to develop and decay (Figs. 7b, 8a, inset). Exponential fits to deactivating tail currents during voltage steps to −120 mV were extrapolated back to obtain the current at the beginning of the voltage change, and this value was then used to determine the fraction of non-inactivated channels for each mutant in the absence of ML277 (Fig. 7e). WT KCNQ1 channels were found to be >60% inactivated, S338A channels were 80% inactivated, and all the mutants at G272 except G272A showed no hook and therefore appeared not to fast inactivate at all. In the presence of ML277 the hook in the tail currents was abolished in WT and all the mutant constructs such that the ratio of initial tail to the extrapolated fit became 1.0 (Fig. 8b). The ratio in L251A and G272C/T/L/V mutants that did not show a tail current hook in control was unaffected by ML277 and remained at 1.0.

There was a strong correlation ($r = −0.86$, $P = 0.0001$, Spearman rank coefficient) between the calculated extent of inactivation and the ability of ML277 to enhance current amplitude (Fig. 8a). Thus, S338A shows augmented inactivation in control and is most responsive to ML277 in terms of current increase, while G272C/T/L/V mutants which don't inactivate do not show any increase in current when exposed to ML277. There are, though, qualitative differences in response. While ML277 increases S338A tails 2× more than WT, there is little effect on the $V_{1/2}$ of activation (Fig. 7d). In V255A, tails in control are very similar to WT (Fig. 8a, inset), and currents increase the same amount upon exposure to ML277, but there is no change in the $V_{1/2}$ of activation (Fig. 7c–e). S338F retains 25% inactivation, but current is decreased in the presence of ML277. Ala mutations at L262 and L266 inactivate almost as much as WT in control, but currents are increased much less, while changes in the $V_{1/2}$ of activation are greater. F335A still inactivates 40%, but tail currents only increase 2× and the $V_{1/2}$ of activation actually depolarizes in the presence of ML277.

WT KCNQ1 single-channel openings are normally brief and small[32,45], but in the presence of ML277 openings are larger and more stable (Fig. 8c)[32]. KCNQ1 S338A shows a greater degree of inactivation than WT (Fig. 7e, Supplementary Fig. 9c, d) and responds more than WT to ML277 (Fig. 7c). Single-channel recordings of S338A in control show brief, low amplitude openings at the limit of resolution (0.022 ± 0.002 pA) with longer closings than WT (Fig. 8c), but in 1 μM ML277, S338A openings are much more frequent and larger than in control (0.049 ± 0.003 pA, Figs. 8c, d). In contrast to WT and S338A, single G272C KCNQ1 mutant channels, which do not inactivate (Figs. 7b, e, 8a), open to a significantly higher level in control, with bursts of openings that last much longer while closings are short-lived. All-points amplitude histograms show a distinct peak

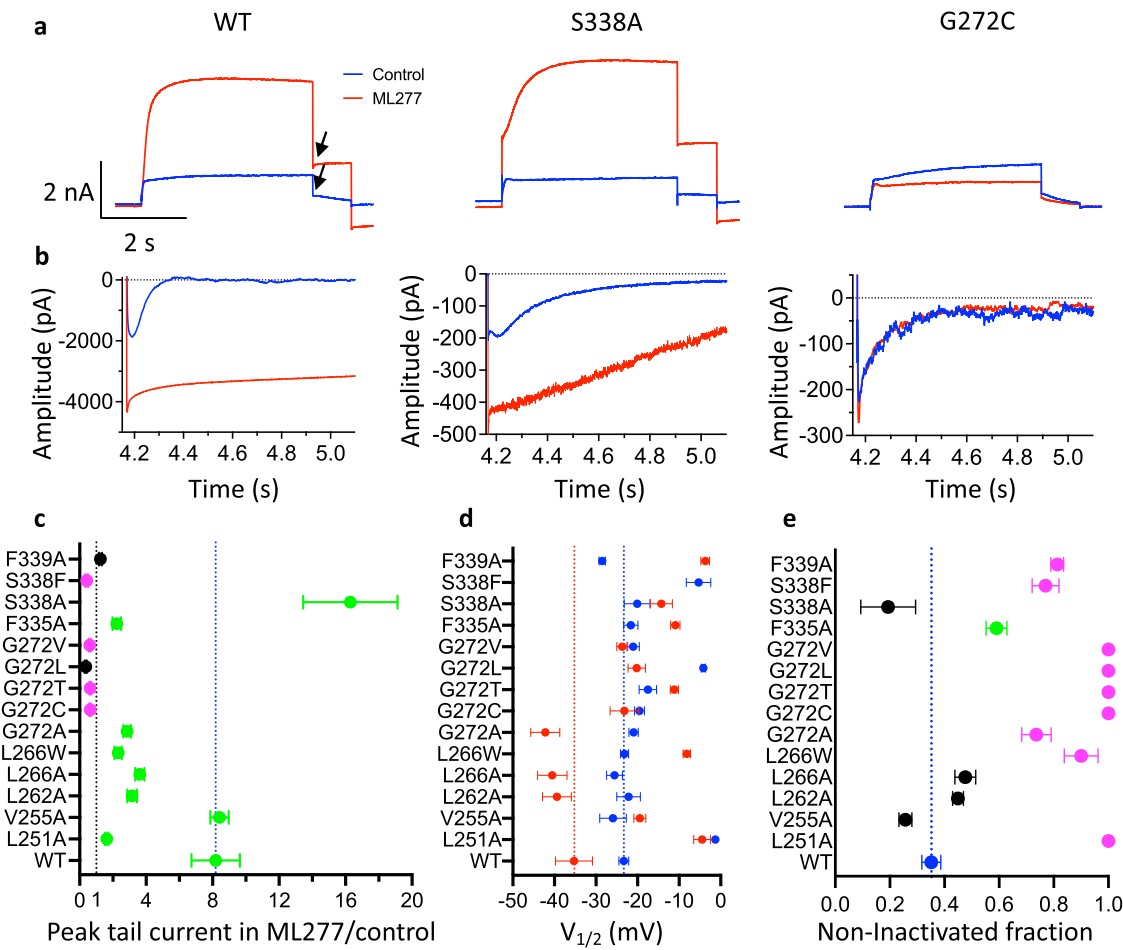

**Fig. 7 Mutations to ML277 binding site residues reduce or eliminate drug effects on hKCNQ1. a** Current traces in control (blue) and after addition of 1 μM ML277 (red) for WT and mutant channels. Cells were held at −90 mV, pulsed to +60 mV for 4 s, then −40 mV for 0.9 s. The interpulse interval was 15 s. Arrows show peak tail measurements for graph in panel **c**. **b** Tail currents at −120 mV after 4 s pulses to +60 mV in control (blue) and in ML277 for different mutants as indicated in **a**. Note: residue numbering in hKCNQ1 is +10 compared with xKCNQ1. **c** KCNQ1 tail current amplitudes in ML277 divided by initial control tail, protocol as in panel **a**. Error bars denote mean ± SEM, $n = 3$–8 cells, (see Supplementary Table 2 for exact $n$ values). Black indicates no drug effect; magenta indicates a significant decrease, and green indicates a significant increase from control (see Supplementary Table 2). Only V255A is not significantly different from WT, otherwise $P = 0.008$–>0.001 using one-way ANOVA. **d** Mean $V_{1/2}$ of activation before (blue) and after (red) exposure to 1 μM ML277. Protocol as in panel **a**. See Supplementary Table 3 for $n$ and mean values and significance tests. For S338F, tail currents in ML277 were too small to measure. **e** Initial tail current amplitude in control divided by extrapolated fit to tail current decay, as a measure of channels that are not inactivated at the start of the tail, values represent the mean ± SEM. WT is blue, not significantly different from WT is black, significantly less inactivated than WT, $P < 0.05$ (green) and $P < 0.01$ (magenta). Where no hook was observed, a value of 1 is given for non-inactivated fraction, e.g., L251A and G272C (**b**). Note that in some cases error bars fall within plotted symbols. $n = 4$ cells for L262A, L266A and F335A, $n = 5$ for L266W, G272V and S338F, for all others $n = 3$ cells. Source data are provided as a Source Data file for panels **c–e**.

for G272C in control (0.040 ± 0.002 pA). This result suggests that the G272C mutant does not respond to ML277 in part because the open state is already more stable than it is in the WT channel.

Taken together, the electrophysiology results described above strongly support the structural data presented in Figs. 1–4, with Leu256, Phe325′, and especially Gly262′ proving key determinants of ML277 activity, and by extrapolation, binding. Removal of inactivation is important to the increase in current induced by ML277, and individual mutations alter the ability of ML277 to interact with different parts of its binding site and thus disrupt different aspects of the overall electrophysiological effect of ML277.

## Discussion

In this report, we have added to the structural understanding of KCNQ1 and KCNQ1/KCNE3 complexed with CaM[4,5] with

additional cryo-EM data to provide a molecular basis for the binding and activity of the KCNQ1 activator, ML277. Two independent data sets revealed an additional density in a hydrophobic pocket formed by the S4-S5 linker helix, the TM-spanning S5 and S6 helices from one protomer, and the S5′ and S6′ helices from an adjacent protomer (Fig. 2). Unlike the major conformational changes associated with channel gating, the binding of ML277 does not trigger major structural changes in comparison to the apo structure and the pore remains in a closed conformation (Supplementary Fig. 3c). Within the binding pocket the changes induced by the presence of ML277 are subtle, see below, and outside this pocket we only observed a small displacement of the C-terminal coiled-coil domain (Fig. 1e, Supplementary Movie 1), again confirmed by refinement of two independent sets of data (Fig. 3).

A single binding location for ML277 has been identified in the present study. Previous experimental and modeling studies have

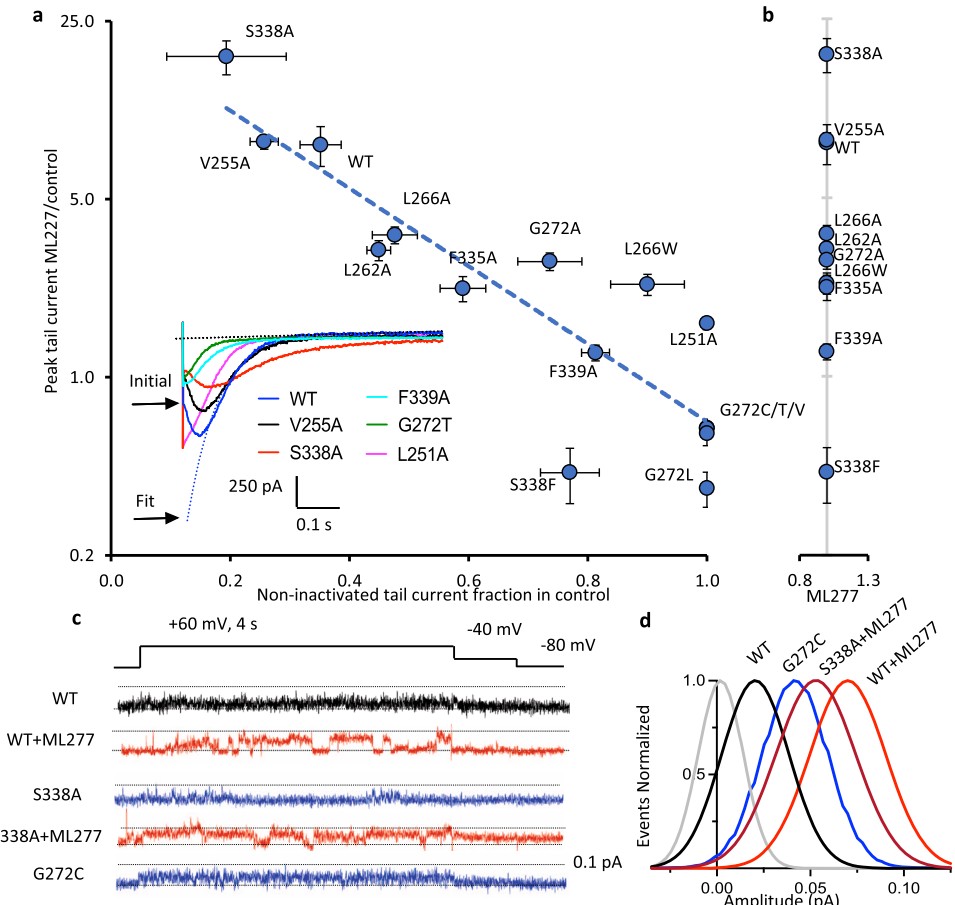

**Fig. 8 Tail current increase in ML277 vs. inactivation of KCNQ1 mutants is reflected in single channels. a** Tail current response to ML277, data from Fig. 7c, on a log ordinate vs non-inactivated tail current fraction in the absence of ML277 (see inset) from Fig. 7e. Fitted correlation $r_s = -0.86$, P = 0.0001, Spearman rank coefficient. Linear trend line is shown. Inset panel shows control tail currents at −120 mV for WT and stated mutants. As an example, the initial peak of the WT tail (blue) is indicated, and also the fit back to the tail start to extract the fitted peak value. Values represent the mean ± SEM, $n$ values as for Fig. 7c and e. **b** As for panel a except non-inactivated fraction was obtained in the presence of 1 µM ML277. L251A and G272C/T/L/V were unchanged from control solution (panel **a**) and for clarity are not replotted. Note that in some cases vertical and horizontal error bars fall within plotted symbols. **c** Voltage protocol and representative single-channel recordings of WT, S338A, and G272C KCNQ1 in control and 1 µM ML277 as indicated. Data filtered at 2 kHz at acquisition, and 200 Hz for presentation. Dotted lines denote zero pA (baseline) and 0.1 pA. No ML277 data were obtained for G272C. **d** Gaussian fits of amplitude event distributions for a blank sweep (gray), WT, S338A, and G272C sweeps, in control and ML277 as indicated. For WT, peaks were 0.02 in control, and 0.069 ± 0.003 pA in ML277, $n = 4$ cells. For S338A, peaks were 0.022 ± 0.002 in control (not shown) and 0.049 ± 0.003 pA in ML277, $n = 3$ cells. For G272C peaks were 0.040 ± 0.002 pA in control, $n = 3$ cells.

suggested various locations on KCNQ1 where ML277 may bind. An earlier study using MD simulations[31] proposed that the drug binds either to an intracellular peripheral site at the boundary of the S2-S3 loop and S4-S5 helix linker of KCNQ1 (R1), or a site between the S5 and S6 helices (R2). It was suggested that ML277 could only access R2 through the R1 site, and that both sites could not be occupied at the same time. Critical residues included F339 (from R2) and F335—setting hKCNQ1 apart from KCNQ2 (corresponding residues Phe304 and Ile300, respectively), an isoform that ML277 is far less active on. It was also suggested that the binding of ML277 likely induced global motions of the channel, first of all restricting pore motions, and secondly involving substructures central to KCNQ1 gating. Our studies do not support such conclusions in so far as they apply to the closed channel with activated voltage sensors, as only a single binding pocket was identified with limited differences between the xKCNQ1-CaM and xKCNQ1-CaM-ML277 structures (Figs. 1, 2). This is similar to the structures of retigabine and linopirdine bound to KCNQ4 where activator or inhibitor binding did not induce any major changes in channel conformation[37]. However,

our experiments do not exclude secondary less stable binding sites for ML277 not visualized in our cryo-EM data or hidden by detergent molecules (Supplementary Fig. 7).

A more recent experimental and theoretical docking study in hKCNQ1 has suggested ML277 binding at the S4-S5 linker-pore interface and identified 13 potential interacting residues in the S4-S5L, the S5 helix, and the S6 helix, including W248R and S338F which abolished the ML277-induced current increase, with W248R also reducing the $V_{1/2}$ shift caused by ML277[18]. Our structural data from two separate protein preparations (data sets 1 and 2, shown in Supplementary Figs. 3–4) indicated the presence of only a single density consistent with the size and shape of ML277 (Figs. 1, 2). The subsequent analysis of excluded 2D classes from xKCNQ1-CaM, xKCNQ1-CaM-ML277 data set 1 (Supplementary Fig. 6), and the direct comparison and overlay of the unmodeled densities from xKCNQ1-CaM and xKCNQ1-CaM-Ml277 also did not suggest alternative locations where ML277 might be present (Supplementary Fig. 7). We emphasize, though, that this does not exclude the presence of other ML277 binding poses in images we did not capture, or during real world

scenarios of activation, inactivation, and deactivation gating. There is no suggestion from electrophysiological studies that ML277 unbinds during the open-closed-open activation cycle, so we take some comfort that the binding pocket that we have identified does not change much from the closed xKCNQ1-CaM structure to the open hKCNQ1-CaM-PIP2 structure as shown by the very similar binding energies that we calculate (Fig. 4). Nevertheless, many gating conformations are traversed during opening and closing pathways about which we have no information.

The ML277 binding pocket identified on xKCNQ1-CaM is lined by residues Trp238, Leu241, and Val245 in the S4-S5 linker; Leu252, Thr255, and Leu256 in helix S5; Pro333 and Leu337 in helix S6 from one protomer and residues Ile258′, Leu261′, Gly262′ and Phe265′ in helix S5′ and Phe322′, Val324′, Phe325′, Ala326′, Ser328′ and Phe329′ in helix S6′ from the neighboring protomer (Fig. 2d). These findings are well supported by the mutational studies described above and our own experimental analysis of hKCNQ1 (Fig. 7). The most effective point mutations that prevented the electrophysiological actions of ML277 were made at G272, as none of the G272C, G272T, G272L, and G272V mutants showed increased current, changes in $V_{1/2}$, or changes in inactivation when exposed to ML277 (Fig. 7, Supplementary Fig. 9). This is an interesting finding as both of the mutations G272D and G272V have been implicated in pathologically significant LQTS1 and sensorineural deafness[39].

The hydrophobic interactions between the methyl-phenyl group of ML277 and the S5′ helix residues Gly262′ and Phe265′ appear critical to its stability of binding and mechanism of action. Mutants of *Xenopus* Leu252 and Leu256 in the S5, (human L262A and L266A or L266W) were quite good at preventing the tail current increase with ML277 (Fig. 7c), but relatively ineffective on the change in $V_{1/2}$, or in preventing the modulation of inactivation. This suggests that residues in helix S5 are less critical determinants of the efficacy of ML277 despite the proximity of the methoxy-phenyl group to helix S5 in this orientation (Fig. 2d). Mutations of residues Phe325′ and Phe329′ in the S6′ of the adjacent protomer showed intermediate efficacy at disrupting the electrophysiological actions of ML277, with loss of current increase (F335A, S338F, F339A in hKCNQ1), with no residual inactivation in the presence of ML277 (S338F, F339A), and prevention of the hyperpolarizing action of ML277 on the $V_{1/2}$ (F335A, S338A, S338F, F339A). Again, mutations S338F, and F339S are implicated pathologically in the development of autosomal dominant LQTS1[46].

S338A proved to be a gain-of-function mutant with equivalent doses of ML277 increasing tail currents twice as much as in WT, while S338A showed increased and more stable inactivation in control as indicated by the hooked tail currents and extremely slow deactivation rate (Figs. 7b, 8a). It is possible that the replacement of one hydroxyl group with a methyl moiety, increasing the local hydrophobicity at this site, allows deeper access into, and higher affinity of ML277 for its binding pocket, therefore stabilizing its activity and increasing its potency (Ser328′ in xKCNQ1 in Fig. 2d). Alternatively, the greater effect of ML277 may simply reflect the fact that this S338A mutant undergoes a deeper and more profound inactivation in control, and reversal of this inactivation upon exposure to ML277 then has a much greater apparent effect.

Identification of a binding pocket for ML277 on KCNQ1 allowed comparison with the recently described binding of ML213 and retigabine to KCNQ4-CaM and KCNQ2-CaM[36,37]. Our study highlights the importance of Gly262′, Phe325′, and Leu256 on xKCNQ1-CaM in not only forming a binding pocket for ML277, but also creating steric clashes that would inhibit the binding of ML213 and retigabine to KCNQ1-CaM (Fig. 6). Our

docking analyses indicated that residues in KCNQ2-CaM and KCNQ4-CaM greatly reduced the binding ability of ML277 to these subfamily members. The presence of W236 and W242 in KCNQ2 and KCNQ4, respectively, prevented access of ML277 to its binding pocket behind the S5. As well, replacement of Gly262′ with C242′ or V248′ provide an obstacle to the binding of the 4-methylphenysulfonamide group of ML277 (Fig. 5).

To understand the actions of ML277 on the open human KCNQ1-CaM channel, in the presence of PIP2, and in the absence of KCNE3, MD simulations were carried out on the open hKCNQ1-CaM-PIP2 (PDB-ID: 6V01) structure[5]. The structures in Fig. 4 suggested that ML277 differs from the action of PIP2 in producing only minor conformational changes in the channel when activating the channel, at least under the conditions used for this cryo-EM study. There were only minor changes to the identified ML277 binding site, and ML277 binding energies between the closed and open conformations of xKCNQ1-CaM-ML277 and hKCNQ1-CaM-PIP2-ML277 (Fig. 4) were similar ($\Delta G$ values of ~−50 kcal/mol, Supplementary Fig. 10). ML277, on binding to the activated-closed (AC) state of xKCNQ1-CaM induces three subtle changes, an enhanced twist in the C-Helix, displacement of the Leu256 side chain, and rotation of the Phe265′ side chain (Figs. 3, 2d). Given these relatively minor structural changes in xKCNQ1-CaM, and likely the open channel as well upon binding of ML277, it is important to understand the large electrophysiological changes that result (Fig. 7).

ML277 interactions bridge several key regions for electromechanical coupling (EMC) and pore gating of the channel in addition to interactions with residues that make contact with S4 in the VSD (L239, Supplementary Fig. 10). To maintain the efficacy and potency of ML277, the length and characteristics of the molecule ends do seem to be necessary[21], suggesting that an ability to bridge the entire region is important. ML277 may enhance EMC by lowering the energy barrier for entering the Activated-Open (AO) state of the channel, through influence on residues/interactions implicated in more traditional EMC mechanisms involving PIP2, the S4-S5 linker and lower S5 and S6 residues, as postulated by others[33]. ML277 interacts with Trp238, Leu241, Val245, Leu252, Ile258, Pro333, Leu337, and Phe329′, important for EMC in KCNQ1[33,47]. Previous studies have shown that W248 is particularly sensitive to mutation with both Ala and Arg resulting in large positive shifts in $V_{1/2}$, and W248R at least, is insensitive to ML277[18,48,49], while in our experiments, mutation of L251 in the S4-S5L prevented all actions of ML277.

Alternate pathways for EMC in Kv channels[50], and perhaps the dominant pathway for non-domain-swapped channels such as hERG[51], involve transmission from S4 directly across to S5 of an adjacent subunit. In the AC state, ML277 interacts in xKCNQ1-CaM with at least one residue in the region identified as important in this alternate mode, Phe265′ (L409 in Shaker channels), which interacts with Ile225 and Leu226 in S4. In the AO state, hKCNQ1-CaM-PIP2 structure, ML277 interacts with L271′ and F275′ in addition to the S4 residue L239 (Supplementary Fig. 10). Three interactions between the S4 and S5 gating interface of Shaker showed higher stability in the activated state[50], which would both facilitate channel opening and have to be overcome for the channel to deactivate. For KCNQ1, MD simulations have suggested that F275 increases the number of contacts with S4 in the AO state[52]. If ML277, in addition to facilitating energy transfer to the pore domain, further stabilizes interactions between S4 and S5, then deactivation rates could also be affected. Mutation of F275 to an alanine shifts the voltage dependence of hKCNQ1 activation to −38 mV[48], and when mutated to a tryptophan the $V_{1/2}$ is shifted positive to ~0 mV[53], indicating that side chain size at this location has significant effects on the $V_{1/2}$ and kinetics.

The data from S338A also support this mechanism, as the mutation, by allowing ML277 to bind deeper into the pocket adjacent the inner pore, could disrupt some interaction with the S4 and explain the faster deactivation of S338A than WT in the presence of ML277 (Fig. 7b).

Finally, the G272C mutant eliminates all electrophysiological actions of ML277 on hKCNQ1, which suggests that the interaction of the drug with the S5′ helix is critical to allow ML277 to enhance EMC or stabilize the open state. Indeed, C242′ in the ML277-insensitive KCNQ2 isoform effectively prevents interaction of the drug with the S5′ helix in KCNQ2 (Fig. 5c, e).

The carboxy terminus of KCNQ1 is fairly long (~307 aa) and contains 4 α-helices, A–D. Helices A and B bind CaM and helices C and D are thought to be important for assembly and trafficking of the channel complex as well as in the case of Helix D, for interaction with AKAPs[54]. Though some contact takes place between the C-helix and CaM in all of the structures. The C-helix is highly conserved, evolving at a rate only slightly higher than the filter region[55], suggesting an important functional role. Several C-helix mutants have been characterized which showed decreased surface expression but some of these mutants also show changes in voltage dependence[56,57], suggesting that proper folding/interaction of this region is required for normal gating of the channel. Our structures are resolved to the human equivalent of aa 566, encompassing the entire C-helix with the remaining truncated C-terminus unresolved (aa 567–610). It is this C-helix region that appears to be more twisted in the presence of ML277 (Figs. 1e, 3, Supplementary Movie 1) than in our and the published apo versions[4]. How binding of ML277 in the transmembrane segment leads to this twist is not obvious, and may involve minor rearrangements not visible at the current resolution. However, alignment of the activated and open inner gate structure of human KCNQ1 (paired with KCNE3, PDB-ID: 6V01[5]) with our apo and ML277 bound structures shows that in order to account for the C-terminal rearrangements necessary for channel opening, this C-helix must move up towards the membrane as the upper half of the helix transitions to a poorly resolved flexible structure. The twist observed when ML277 binds may be related to the CTD's final conformation in the open channel and perhaps this pre-twist may prime the channel to open more readily, or close less easily.

Several mutations of hKCNQ1 in the binding pocket for ML277 inherently abolish (L251A), or greatly reduce (L266W) fast inactivation in KCNQ1. The G272C/T/L and V mutants and L251A lack a hook upon repolarization, and are insensitive to the activator effects of ML277 (Fig. 7). This suggests the importance of G272′ in the process, and that the binding of ML277 to this area could restrict the mobility of the region surrounding G272′, including L271′ and F332′. In xKCNQ1, Gly262′ is located immediately next to the side chain of Phe322′ (Fig. 2d) and bulkier residues at position 262′ are very likely to affect the conformation and dynamic behavior of the Phe322′ side chain. Flipping of Phe322′ also is not possible without some rearrangement of the S5′ helix, which may be facilitated by Gly262′ allowing more main chain flexibility.

There was some ambiguity surrounding the assignment of Phe322′ from the xKCNQ1-CaM density maps with two orientations possible (Fig. 9). In hKCNQ1, F332′ seems clearly to point towards the filter helix (PDB-ID: 6UZZ and 6V01[5]), while in the prior xKCNQ1 structure Phe322′ is modeled towards the helix but the density map is more ambiguous, though the structure itself is of lower resolution (PDB-ID: 5VMS[4]). With Phe322′ sitting behind the filter helix and interacting with residues in the helix, the ability to flip conformation could contribute to a filter gate mechanism, and perhaps underpin the unique flicker appearance of KCNQ1 single-channel openings[32,45] and

inactivation. Although more data are required to understand the mechanism, these observations suggest that ML277 may disrupt fast inactivation by immobilizing critical residues involved in the inactivation process.

In support of this notion, the mutation F332A produces non-inactivating channels largely insensitive to ML277[31,58]. The co-assembly with KCNE1, known to eliminate inactivation in KCNQ1, also makes the channel resistant to the effects of ML277[14], although this is not true of KCNE5, which does not eliminate inactivation in KCNQ1[58] and responds to ML277 with a doubling of current[32]. As well, the phenylalanines equivalent to Phe322′ in KCNQ2 and KCNQ4 (F297 in KCNQ2 and F303 in KCNQ4) which do not fast inactivate[59,60], point away from the filter helix with little ambiguity in the density maps[36–38]. Each of these channels contains a leucine in the filter helix that would, for steric reasons, hinder phenylalanine from flipping towards the helix (L275 in KCNQ2 and L281 in KCNQ4). These channels produce clear openings of larger amplitude under control conditions[61,62], similar to KCNQ1 in the presence of ML277[32], and quite unlike the small amplitude and flickery nature of KCNQ1 openings in the absence of ML277 (Fig. 8).

This work provides one definitive location for an ML277 binding pocket in KCNQ1 channels. Mutations of residues lining the binding pocket strongly support the described location for the ML277 cryo-EM density bounded by the S4-S5 linker and the S5 and S6 helices from one subunit, and the S5′ and S6′ helices of a second adjacent subunit. It is clear that prevention of channel inactivation is an important component of the ML277 action as none of the mutants that we tested, that no longer inactivated at all, responded to ML277 (G272C/L/T/V and L251A). Data suggest that IKs in myocytes may not be fully saturated with KCNE1[14,31], and if this is also true of IKs in the intact human heart, or perhaps even more importantly, the diseased heart, ML277, or derivatives, have potential use as therapeutic agents to treat patients with LQTS1. Our description of a principal binding site will pave the way for a better understanding of how putative drugs can rescue the channel dysfunction characteristic to LQTS1, and potentially open avenues for the development of drugs structurally and functionally similar to IKs activators like ML277.

## Methods

**Cloning and expression of xKCNQ1/CaM.** A tagged *Xenopus laevis* KCNQ1 (NP_001116347.1) construct, encoding residues 67–610 of the channel, was ordered from Genewiz (South Plainfield, NJ, USA). The truncation of the N- and C-termini was to produce a more stable protein construct as established by Sun and MacKinnon, 2017[4]. To allow for channel complex purification, we also opted for the same C-terminal green fluorescent protein (GFP)−6xHis tag and a preScission protease site for later excision of the tag. DNA encoding calmodulin (CaM), which is absolutely conserved from frog to human, was obtained as a kind gift from the Wayne Chen lab (Calgary, Canada). Both KCNQ1 and CaM constructs were cloned into the pcDNA3 expression vector.

Before transfection, $87.5 \times 10^6$ tsA201 adherent cells (with the density of $10^5/cm^2$) were incubated for 24 h in minimum essential media (MEM) supplemented with 10% (v/v) fetal bovine serum at 37 °C. DNA samples were co-transfected into the tsA201 cells at a ratio of KCNQ1:CaM 5:1 using polyethylenimine (PEI) transfection reagent, which was previously found an efficient strategy for expression in mammalian cell lines[32,63]. The transfected cells were harvested 48 h post-transfection and KCNQ1-CaM was subsequently purified.

**Purification of xKCNQ1-CaM using an anti-GFP nanobody-based affinity purification.** Resins coupled with high-affinity anti-GFP nanobody (NB) were utilized for purification[64]. The NB resin was prepared using purified 6xHis-TRX-NB. The NB protein was expressed in Rosetta gami 2 cells which were incubated at 30 °C until an optical density at 600 nm ($OD_{600}$) of ~0.8 was reached, grown overnight at 25 °C upon induction with 250 μM IPTG. The cells were suspended in lysis buffer (50 mM HEPES pH 7.4, 300 mM NaCl, 10% w/v sucrose, lysozyme, and DNase) and sonicated for 2 min (1 s on/off) at 50% amplitude, twice. The suspension was spun down using a JA25.5 fixed-angle rotor (Beckman) at $39,000 \times g$ for 40 min, the supernatant was filtered through a 0.45 μm filter and the NB was

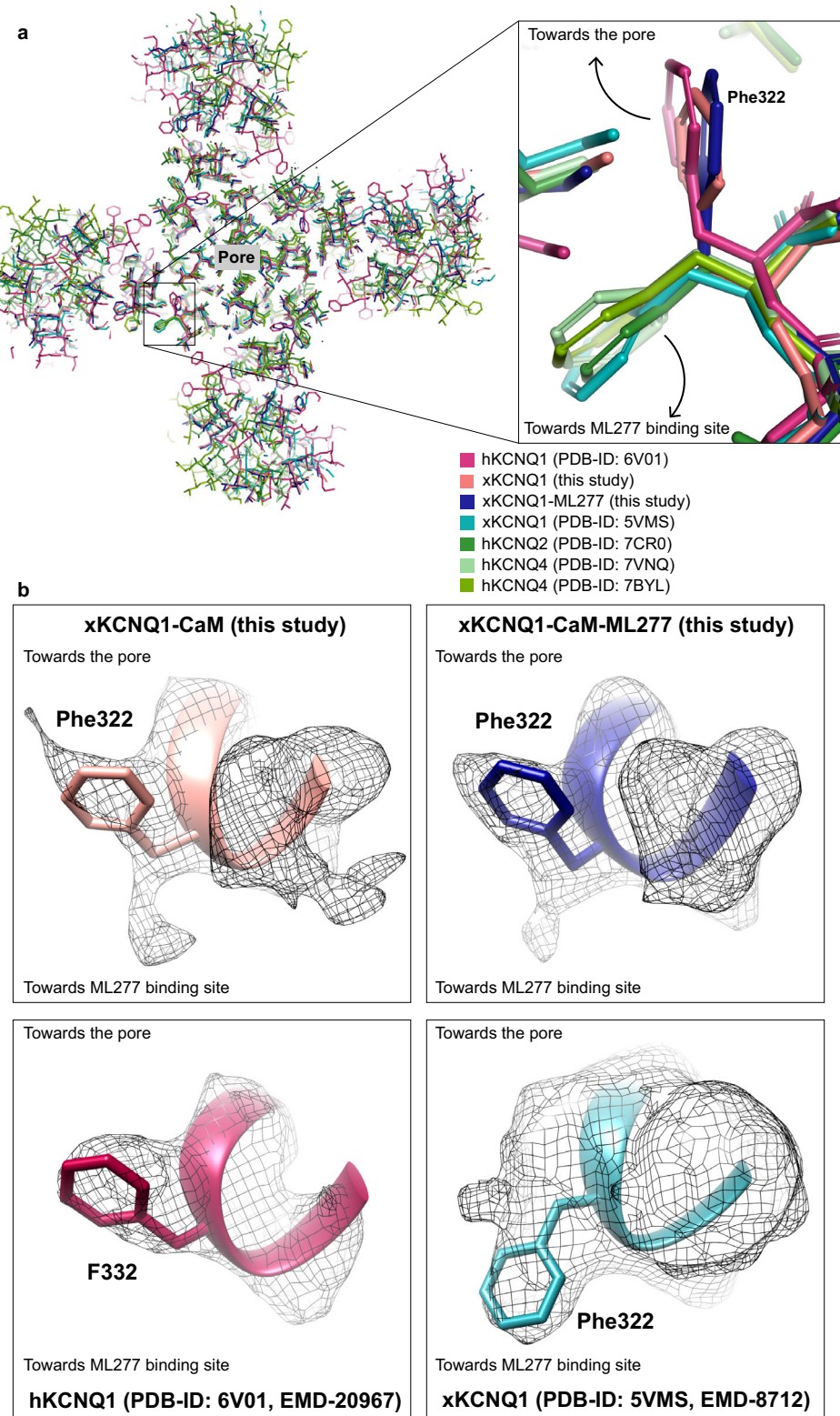

**Fig. 9 Conformation of Phe322 in xKCNQ1-CaM. a** Superposition of xKCNQ1-CaM and xKCNQ1-CaM-ML277 with hKCNQ1 (PDB-ID: 6V01), xKCNQ1 (PDB-ID: 5VMS), hKCNQ2 (PDB-ID: 7CR0) and hKCNQ4 (PDB-ID: 7VNQ, PDB-ID: 7BYL) structures showing detail around Phe322 (xKCNQ1) in a top view from the extracellular side. The previously published xKCNQ1-CaM and hKCNQ2 and hKCNQ4 structures show a corresponding Phe322 side-chain orientation pointing towards the ML277 binding site, while hKCNQ1 and our structures show a Phe322 oriented toward the pore. **b** Cryo-EM densities for Phe322 in xKCNQ1-CaM and xKCNQ1-CaM-ML277 (above) from this study, and from hKCNQ1 (F332, PDB-ID: 6V01) and xKCNQ1-CaM (Phe322, PDB-ID: 5VMS), below. The contour level cut offs were: maps from this study 8x rmsd, EMD-20967 6x rmsd, EMD-8712 1x rmsd.

purified using affinity and size exclusion chromatography columns applied on FPLC (ÄKTA, Cytiva). First, the protein solution was loaded on a HisTrap FF column (Cytiva) which was pre-equilibrated in binding buffer (containing 50 mM HEPES pH 7.4 and 300 mM NaCl). Next, the column was washed with 5 % elution buffer (50 mM HEPES pH 7.4, 300 mM NaCl, and 500 mM imidazole) and bound proteins were eluted with 100% elution buffer. 6xHis-TRX-NB was dialyzed in dialysis buffer (50 mM NaPi buffer pH 7.5 and 300 mM NaCl). Tobacco Etch Virus (TEV) protease was added to the sample prior to dialysis to cleave the 6xHis-TRX-tag. TEV protease and uncleaved protein were separated from the cleaved NB using a HisTrap FF column (Cytiva) following the same procedure as described above using 50 mM NaPi buffer pH 7.5 instead of HEPES. The protein was subsequently concentrated and loaded on a Superdex 75 increase 10/300 GL (Cytiva) in dialysis buffer. The eluted protein was concentrated to 1 mg/mL and directly used for the preparation of the NB resin.

For the preparation of the NB resin, 15 mL NHS-activated Sepharose 4 FF (Cytiva) was washed with 250 mL ice-cold 1 mM HCl and then equilibrated with 150 mL of dialysis buffer in three increments. For coupling, ~100 mg of NB (at a concentration of about 1 mg/mL) was added to the resin and incubated overnight at 4 °C while stirring. The resin was exchanged to fixation buffer (100 mM Tris pH 8) for 4 h at 4 °C and washed with cold fixation buffer in three increments of 50 mL. The NB-resin was stored in 100 mM Tris pH 7.4 at 4 °C for over one year.

For purification of xKCNQ1, a pellet from ~2–4 × 10⁹ cells was suspended in 50 mL lysis buffer (40 mM HEPES pH 7.2, 150 mM KCl, 0.5 mM CaCl₂, benzonase, 2 mM MgCl₂ and 1× protease inhibitor cocktail set III, EDTA-Free). The xKCNQ1-CaM suspension was then solubilized through the addition of a detergent mixture DDM/CHS (5:1 ratio, 6% w/v DDM) to obtain a final concentration of 1% (w/v) DDM, while stirring at 4 °C for 45 min. Insoluble material was removed using high-speed centrifugation ($100,000 \times g$ for 40 min at 4 °C), and the supernatant was incubated with 2 mL of the NB-resin under rotation at 4 °C for 4 h to allow xKCNQ1-CaM binding. The xKCNQ1-CaM was eluted through PreScission protease cleavage overnight at 4 °C. The xKCNQ1-CaM complex and PreScission were separated by centrifugation of the resin. PreScission protease was removed from the xKCNQ1-CaM sample using glutathione sepharose beads (Cytiva) and purified KCNQ1-CaM was collected and concentrated to 1–2 mg/mL using a 100-kDa NMWL centrifugal filter unit (Amicon Ultra). The concentrated sample was immediately used for cryo-EM grid preparation.

**Cryo-EM sample preparation, data collection, and analysis.** ML277 (solubilized in 100% v/v DMSO at 50 mM) was added to a solution containing 1 mg/mL of purified KCNQ1/CaM complex, to a final concentration of 100 µM ML277 (resulting in 0.2% v/v final DMSO concentration) 20 min prior to grid preparation. Quantifoil R1.2/1.3 copper grids with holey carbon (Electron Microscopy Sciences) were glow-discharged (PELCO easiGlow). Two microliters of xKCNQ1-CaM or xKCNQ1-CaM-ML277 was applied to the grids and plunge frozen using a Mark IV Vitrobot (ThermoFisher Scientific) at a blot force of −10 to −15, 4 °C for 3 s. Grids were prescreened on a Glacios microscope (ThermoFisher Scientific) equipped with a Falcon III detector, and high-quality grids were used for data-collection on a Titan Krios (ThermoFisher Scientific) at the HRMEM facility at UBC.

Two data sets were collected for xKCNQ1-CaM-ML277. The first data set was composed of 4,403 micrographs on a Falcon III direct electron detector (ThermoFisher Scientific) with a resolution pixel size corresponding to 0.85 Å. Here, 48 total frames were obtained following a total dose of 50 elections per Å². The second data set was composed of 13,572 micrographs collected using a Falcon 4i Direct Electron Detector (ThermoFisher Scientific) in super-resolution mode with a binned pixel size of 0.77 Å, and a total dose of 50 elections per Å². All data sets were collected at a defocus range between −0.5–3.0 µm. The final data set of xKCNQ1-CaM included 19,997 micrographs collected on the Falcon 4i Direct Electron Detector in super resolution at a binned pixel size of 0.77 Å, and a total dose of 50 elections per Å².

For the xKCNQ1-CaM data set, super-resolution gain-corrected images were 2 x binned (final pixel size 0.77 Å). Except when indicated differently, all processing was done using cryoSPARC 3.3.1[65]. Movies were patch motion corrected using all frames. Defocus values were calculated using CTFFIND4[66]. Processing was performed with respective particle box sizes of 380 for xKCNQ1-CaM and xKCNQ1-CaM-ML277 (data set 2) and 360 pixels for xKCNQ1-CaM-ML277 (data set 1). CrYOLO 1.7.6[67] was used for particle picking of all data sets, where training was done on 15–20 micrographs. Initial picking led to 1.83 million, 496,000, and 531,573 particles for the xKCNQ1-CaM, xKCNQ1-CaM-ML277 (data set 1), and xKCNQ1-CaM-ML277 (data set 2), respectively. Clean-up of the particle sets was done by one round of 2D classification. For xKCNQ1-CaM, micrographs were curated to have a CTF fit better than 5.0 Å and the resulting 71,241 particles were used for ab initio 3D reconstruction and non-uniform refinement[68] to obtain a final reconstruction of 3.8 Å resolution. For xKCNQ1-CaM-ML277 (data set 1), the cleaned up 32,900 particles after 2D classification was used for ab initio 3D reconstruction and subsequent non-uniform refinement to obtain a final reconstruction of 3.97 Å. The particles were further classified through heterogenous refinement into four classes. Particles from two similar classes, accounting for 56.8% of particles, were pooled and further refined using non-uniform refinement to 4.8 Å, while a third class with 41.3% of particles was refined to 4.1 Å resolution. For xKCNQ1-CaM-ML277 (data set 2), the cleaned up 85,400 particles after 2D

classification were used for ab initio 3D reconstruction with two classes, which further cleaned up the particle stack, with 62.5% contributing to a significant KCNQ1 class. Particles from this class were sequentially, homogeneously, and non-uniformly refined to a final reconstruction of 4.8 Å. The cryo-EM data collection, refinement, and validation statistics for xKCNQ1-CaM and xKCNQ1-CaM-ML277 (data set 1) may be found in Supplementary Table 4.

The xKCNQ1-CaM and xKCNQ1-CaM-ML277 models were built in Coot[69] using published xKCNQ1-CaM (PDB ID: 5VMS)[4] as a starting model. The xKCNQ1 model was built from residues 95 to 556, with disordered regions 67–94, 206–214, 385–496 missing. Calmodulin was built from residues 10 to 146. Potassium ions were not modeled inside the selectivity filter due to limited resolution. The models went through several rounds of real space refinement with secondary structure restraints using PHENIX[70]. Model quality was assessed using Molprobity[71]. Figures were generated using PyMOL (Schrodinger), Chimera[72], ChimeraX[73], and HOLE[74].

**Molecular dynamics.** We performed MD simulations for KCNQ1-ML277 and KCNQ1-PIP2-ML277, which were built using the structure of the human KCNQ1-CaM-KCNE3 complex with PIP2 (PDB-ID: 6V01). We built the KCNQ1-PIP2-ML277 complex using the superimposed coordinates of open-state hKCNQ1 in complex with PIP2, and ML277 from xKCNQ1-CaM-ML277. Then, ML277 and its vicinity were optimized using ICM-Pro[44], sampling side chains conformations to find the energetically most favorable state and minimize the energy of the protein-ligand complex. For MD simulations, we used only the pore region with the VSD, and the tetramerization domain was trimmed. For xKCNQ1-ML277 we used residues Asn95-Gln349, and for hKCNQ1-PIP2-ML277 residues Thr104-Val355. The complexes were inserted into the POPC lipid belayer and solvated using TIP3P water models with 150 mM K⁺/Cl⁻ ions[75]. The system was assembled using CHARMM-GUI, and MD simulations were run with GROMACS 2021.4[76–78]. We used the CHARMM36m (Jul 2021 release) force field for protein, lipids, ions, and CGenFF for ML277 parametrizations[79–81]. Dihedrals of ML277 with high penalty scores were parametrized using CGenFF-optimizer with MP2/6–31G(d) QM calculations performed in PSI4[82–84]. We minimized and equilibrated the assembled systems, gradually releasing restraints from backbone and side chains, and performed a final 10 ns equilibration without restraints. In the case of hKCNQ1-PIP2-ML277, during the last 10 ns equilibration, only ML277 was restrained. Finally, we carried out 200 ns simulations at a constant temperature of 300 K and pressure of 1 bar, controlled with the v-rescale thermostat and the semi-isotropic Parrinello-Rahman barostat respectively[85,86]. Long-range electrostatic forces were treated with the particle-mesh Ewald (PME) summation, and Lennard-Jones interactions were cut off at 1.2 nm with a force-switch modifier from 0.8 to 1.2 nm[87]. We constrained hydrogen-containing bonds with the LINCS algorithm and used a 2 fs integration time step[88].

**Free energy calculation.** We performed binding free energy calculations using the MMPBSA method (molecular mechanics energies combined with the Poisson–Boltzmann surface area) with an implicit membrane model. Topology files were converted from GROMACS to AMBER format with ParmEd, and calculations were performed with the MMPBSA.py program using 200 snapshots collected from the last 100 ns[89].

**Docking.** We used ICM-Pro v3.9 for molecular docking[44]. We added hydrogens of ML277 at neutral pH, optimized covalent geometry, and assigned atomic charges. ML277 neighbor residues at a distance 5 Å were selected as the binding site. In all cases, docking was performed with side chain and ligand flexibility imitation with a Sampling Effort of 50. We considered all rotatable bonds in ML277 and sampled all possible conformations to find the most energetically favorable one which fit the binding site. The Sampling Effort is a parameter in ICM-Pro that represents the thoroughness of the docking.

**Whole-cell and cell-attached patch-clamp experiments.** Whole-cell experiments were carried out using tsA201 cells for the greatest expression, while cell-attached experiments utilized mouse *ltk-* fibroblasts for their lower levels of endogenous K⁺ currents. As previously described[12] cells were plated on coverslips and transfected the next day with Lipofectamine as per the manufacturer's protocol with 2 µg of GFP-tagged WT or mutant KCNQ1 in pcDNA3. Recordings were made 24–48 h after transfection.

Whole-cell currents were recorded using an Axopatch 200B amplifier, Digidata 1440A, and pClamp 10 software (Clampex10.1, Molecular Devices, San Jose, CA, USA). Electrode resistances were between 1–2 MΩ, with series resistances <4 MΩ and compensation of ~80% was applied with a calculated voltage error of ~1 mV/ nA current. Currents were sampled at 10 kHz and filtered at 2–5 kHz[12]. Single-channel recordings were acquired with an Axopatch 200B amplifier, Digidata 1330 A and pClamp 9 software (Molecular Devices, San Jose, CA, USA). After fire polishing, single-channel electrode resistances were between 40 and 60 MΩ. Prior to use, electrodes were coated with Sylgard (Dow Corning, Midland, MI, USA). Records were sampled at 10 kHz, low-pass filtered at 2 kHz at acquisition using a −3 dB, four-pole Bessel filter, and digitally filtered at 200 Hz for presentation and analysis using Clampfit 10.1[12,90,91].

**Electrophysiology solutions**. For whole-cell recordings, the bath solution contained (in mM): 135 NaCl, 5 KCl, 1 MgCl$_2$, 2.8 NaAcetate, 10 HEPES (pH 7.4 with NaOH). The pipette solution contained (in mM): 130 KCl, 5 EGTA, 1 MgCl$_2$, 4 Na$_2$-ATP, 0.1 GTP, 10 HEPES (pH 7.2 with KOH). For single-channel recordings, the bath solution contained (in mM): 135 KCl, 1 MgCl$_2$, 1 CaCl$_2$, 10 HEPES, 10 Dextrose (pH 7.4 with KOH). The pipette solution contained (in mM): 6 NaCl, 129 MES, 1 MgCl$_2$, 5 KCl, 1 CaCl$_2$, 10 HEPES (pH 7.4 with NaOH).

**Analysis**. Conductance-Voltage (G-V) plots were obtained from normalized tail current amplitudes. Tail currents were measured immediately upon repolarization to −40 mV and normalization was performed to the peak initial value achieved during the step to −40 mV for each cell. A Boltzmann sigmoidal equation was used to fit G-Vs (Prism 9, GraphPad Software, San Diego, CA) to obtain the $V_{1/2}$ of activation and slope factor, $k$. Gaussian fits of all-points histograms of single-channel events (using 0.01 pA bin widths) were obtained in Clampfit 10.1 (Molecular Devices, San Jose, CA, USA). All measurements were taken from distinct cells and repeat data samples obtained from the same cells were not used in data summaries. Results are reported as mean ± SEM, unless otherwise stated. Statistical comparison was performed in Prism using either one-way ANOVA, two-tailed Student's $t$-tests or Mann–Whitney tests. $p$-values < 0.05 were considered to be statistically significant.

**Reporting summary**. Further information on research design is available in the Nature Research Reporting Summary linked to this article.

## Data availability

Structural data that support the findings of this study have been deposited in the PDB with the numbers PDB-ID: 7TCP [https://doi.org/10.2210/pdb7TCP/pdb] (xKCNQ1-CaM) and PDB-ID: 7TCI [https://doi.org/10.2210/pdb7TCI/pdb] (xKCNQ1-CaM in complex with ML277). Other previously published PDB files used in this manuscript were PDB-ID: 6V01 [https://doi.org/10.2210/pdb6V01/pdb] (KCNQ1-KCNE3-CaM complex with PIP2); PDB-ID: 5VMS, [https://doi.org/10.2210/pdb5VMS/pdb] (CryoEM structure of Xenopus KCNQ1 channel); PDB-ID: 7CR3, [https://doi.org/10.2210/pdb7CR3/pdb] (human KCNQ2-CaM in apo state); PDB-ID: 7BYL, [https://doi.org/10.2210/pdb7BYL/pdb] (Cryo-EM structure of human KCNQ4), PDB-ID: 6UZZ, [https://doi.org/10.2210/pdb6UZZ/pdb] (human KCNQ1-CaM complex); PDB-ID: 7CR0, [https://doi.org/10.2210/pdb7CR0/pdb] (human KCNQ2 in apo state); PDB-ID: 7CR7, [https://doi.org/10.2210/pdb7CR7/pdb] (human KCNQ2-CaM in complex with retigabine; PDB-ID: 7VNQ: [https://doi.org/10.2210/pdb7VNQ/pdb] (human KCNQ4-ML213 complex); PDB-ID: 7CR4 [https://doi.org/10.2210/pdb7CR4/pdb] (human KCNQ2-CaM in complex with ztz240). All other data supporting the findings of this study are available within the paper, its Supplementary information files as well as a Source data file. Source data are provided with this paper.

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

## Acknowledgements

This research was funded by grants to D.F. from the Natural Sciences and Engineering Research Council of Canada (Grant RGPIN-2016-05422), the Canadian Institutes of Health Research (Grants PJT-156181, and PJT-175024), and the Heart and Stroke Foundation of Canada (Grant G17-0018392). F.V.P. acknowledges funding from the CIHR (PJT 148632). The authors acknowledge Dr. Claire Atkinson and the operating team of the High-Resolution Macromolecular cryo-Electron Microscopy facility (HRMEM) at UBC for data collection. HRMEM is funded by the Canadian Foundation of Innovation, BC Knowledge Development Fund, and the University of British Columbia. The authors acknowledge that this work was carried out on the UBC Point Grey (Vancouver) campus, which sits on the traditional, ancestral, unceded territory of the xʷməθkʷəy̓əm (Musqueam) First Nation.

## Author contributions

K.W., J.E., E.K., F.A., Y.D., and D.F. collected data. H.S. performed MD simulations. K.W., J.E., E.K., D.F., and F.V.P. reviewed, analyzed, and interpreted data. D.F., J.E., E.K., K.W., H.S., S.R., and F.V.P. wrote, edited, and reviewed the manuscript. All authors approved the final version.

## Competing interests

The authors declare no competing interests.
