## [Peer Review File · Nature Communications]

Structural and electrophysiological basis for the modulation of KCNQ1 channel currents by ML277REVIEWER COMMENTS

Reviewer #1 (Remarks to the Author):

The voltage-gated potassium channel KCNQ1 plays important physiological functions, and dysfunction thereof is linked to disease. Therefore, KCNQ1 is a highlighted pharmacological target and mechanistic insights into how KCNQ1 interact with small-molecule compounds is of great interest. In this study, Willegems and co-workers provide Cryo EM structures of KCNQ1 with and without the activator ML277, which allows the authors to propose a ML277 binding site and provide side-by-side comparison of structures without and with ML277 bound. The authors conclude that ML277 binds to a hydrophobic pocket formed by the S4-S5 linker and transmembrane segments 5 and 6, in which ML277 interacts with a large set of residues. In support of this, site-directed mutagenesis and electrophysiology experiments show that mutation of several of the proposed interacting residues alter the channel's response to ML277. Moreover, the authors take advantage of previously determined structures of KCNQ1/KCNE3, KCNQ2, and KCNQ4 to propose structural/mechanistic basis for ML277 selectivity. As such, the authors make several important contributions of relevance to the field. The manuscript is well written, and the extensive work described therein appears to be technically sound. However, given the authors expertise in ion channel biophysics and extensive previous experience with KCNQ1, the authors are encouraged to clarify/develop their mechanistic interpretations. This is particularly pertinent to their interpretation and conclusions related to parameters that determine the channel's response to ML277, but also applies to the putative role of PIP2 and KCNE subunits (see comments below).

General comments:

1) As described in previous work and the present study, ML277 has several activating effects on KCNQ1, such as shifting $V_{1/2}$ of activation, increasing the overall tail current, and preventing channel inactivation. However, it is not evident how ML277 induces these different effects from a single binding site. Could the authors please discuss how they envision ML277 to induce these effects from the binding site identified in this study? In this context, please also comment on the finding that the structure with ML277 bound is in a closed conformation, despite ML277 being a channel activator. As pointed out by the authors themselves (row 184-189), the ML277 binding site might be distorted in other (open) conformations, the details of which the present structure does not provide insights into.

2) The authors provide an impressive amount of electrophysiology data to functionally assess the importance of KCNQ1 residues predicted to interact with ML277. The authors use the pharmacological response of these mutants as a readout for preserved or altered ML277 binding (e.g. row 346-348). At the same time, the authors appreciate the importance of channel inactivation for ML277 effects, for instance as a possible mechanistic basis for lack of ML277 effects on KCNQ1/KCNE1 (which does not inactivate, row 448-454). This raises the question as to what extent altered response of proposed binding site mutants to ML277 is caused by impaired binding (validating the ML277 binding site) or

could be explained by altered inactivation behaviour of mutants (not validating the ML277 binding site). Could the authors please clarify and motivate when they interpret impaired ML277 binding or altered intrinsic inactivation as the cause of smaller effects?

3) In relation to this, please comment on how the authors model aligns with the previously described effect of ML277 on KCNQ1/KCNE1 with less than four KCNE1 subunits in the complex. Is the effect possible due to one or several available ML277 binding sites not occupied by KCNE1 or because of the inactivation behaviour of unsaturated KCNQ1/KCNE1 complexes?

4) The phospholipid PIP2 is required for electromechanical coupling in KCNQ1. The authors describe that the ML277 binding site is anticipated to be distorted upon PIP2 binding (row 250-255). Please discuss the anticipated functional consequence of PIP2 binding in the context of ML277's ability to bind to its site and act as a channel activator.

5) The slightly twisted C-terminal coiled coil domain in the ML277 bound structure compared to unbound structures is highlighted throughout the results section. However, the putative functional relevance of this twist is not commented upon. Please discuss.

Specific comments:

The authors comment upon "additional density, likely representing unmodeled detergent or lipid molecules" in the Figure 1 legend. Is there any chance these densities represent ML277 in alternative sites?

Line 104: "neuronal Kv7 channels" could be "neuronal KCNQ channels" to consistently use KCNQ nomenclature.

Line 147: Please include numerical values (mean +/- sem) for ML277 effects on the truncated xKCNQ1, to allow for quantitative comparison to the effect on human KCNQ1.

Line 182: "kcnq1" should be "KCNQ1".

Line 196: "...hindered sterically...". I find this phrasing hard to understand.

Line 228-229: Could the role of the side chain bulkiness for channel selectivity of listed residues possibly be supported by mutagenesis experiments?

Line 283: Please clarify that this effect is for 1 μ M of ML277.

Line 293-294: This sentence is hard to understand.

Figure 1 legend: Please explain what HA-HD denotes.

Figure 5C-D: Why is not sem included?

Extended Data Figure 1D and 9: Please explain how normalization was done.

Extended Data Figure 1E: is this a single representative recording? Otherwise, please include sem.

Extended Data Figure 2A: "xKNQ1-CaM" should be "xKCNQ1-CaM".

Supplementary Information: The title is not consistent with the title of the main work.

Reviewer #2 (Remarks to the Author):

In the manuscript, Willegems and colleagues describe the binding site of KCNQ1 activator ML277 utilizing cryo-electron microscopy and electrophysiological techniques. The study offers novel and accurate structural insights into how the KCNQ1 activator binds to the channel and how corresponding gating modifications may occur. Comparison of cryo-EM structures obtained with and without ML277 revealed a binding conformation of a drug and explains the isoform specificity of its action. Authors show that the binding pocket of the channel is formed by several residues of S4-S5 linker, S5 and S6 pore helices of one subunit as well as by some residues in S5 and S6 segments of the neighboring subunit. The structures likely represent a configuration of the channel with activated voltage-sensor and closed pore, which in line with what has been reported for KCNQ1 channels without obligatory ligand

PIP2. The absence of PIP2 in the structures makes it difficult for authors to interpret the structural data in context of the precise mechanism of ML277 action. However, authors performed a detailed mutational analysis of the key residues coordinating ML277 revealed from structural data and could show that these residues also play an important role in the drug-induced gating modifications for fully functional channels that likely have PIP2 molecules bound to their structure. The study has important implications for understanding the mechanism of the ML277 action on KCNQ1 channels, which is an imperative step towards the design of a more potent and highly specific analog(s) of the drug for potential treatments of KCNQ1-related pathologies. I only have a few minor comments that I hope will help authors to improve the manuscript:

1) A remarkable modification of wild type homomeric KCNQ1 channel by ML277 is that the drug renders the channel constitutively open. The fraction of constitutively open channels is quite large already at 1 μ M concentration of the drug as electrophysiological experiments indicate. Similar results were also reported earlier by authors. Represented structures, nevertheless, indicate that the inner gate of the channel is closed. This could be due to the fact that PIP2 is missing from the KCNQ1-CAM-ML277 complex. A concise and comprehensive discussion about the putative mechanism of how ML277 may render the channel constitutively open in the context of the presented structural data would be very interesting.

2) Related to the point 1: the extent of ML277-induced current potentiation in S338A mutant is roughly twice as large compared to wild type. However, ML277 does not seem to have a large constitutively active component. The same is true for V255A mutant – increase of tail current is comparable with wild type but no large constitutively open fraction is observed. What is the explanation for these discrepancies?

3) In the manuscript (lines 238-240) authors write “... ML277 as a KCNQ1 activator does not mimic the action of PIP2 on the channel conformation, at least under the conditions used for this cryo-EM study”. For readers it is unclear why authors expect ML277 to mimic the action of PIP2 molecules. To my knowledge, no functional data are so far reported that show an effect of the drug that resembles PIP2-like action on KCNQ1 channels. In addition, the reported amino acids that coordinate PIP2 in KCNQ1, with exception of W248, are different from those that coordinate ML277. This section could be improved so that the framework of the arguments become clearer to the readers.

4) The normalized peak tail current shown in the extended data (Fig. 9) for L251A mutant does not correspond to the peak currents shown in the right panel. This is evident when the current levels corresponding to potentials more negative than -20 mV are considered.

Reviewer #3 (Remarks to the Author):

This study by Willegems et al reports cryoEM structures of *Xenopus* KCNQ1 associated with Ca-calmodulin in the absence (xQ1-CaM) or presence of ML277 (xQ1-CaM-ML277) (Figs. 1 and 2). The authors tested whether mutations in the putative ML277 binding site (made in the human KCNQ1 ortholog) could alter ML277's effects on this channel (Fig. 5A-C, supplemental Fig. 9). They used these functional data, as well as some single channel data (Fig. 6), to support the statement that ML277 increased KCNQ1 current amplitude by removing KCNQ1 inactivation. Finally, the authors extensively discussed why ML277 is relatively specific toward KCNQ1 vs the other members of the KCNQ family, and how PIP2 and KCNE3 can affect ML277 binding to KCNQ1 (Figs. 3 and 4). This latter part is solely based on alignments of their structure(s) with published cryoEM structures, instead of any quantitative analysis, such as calculation of *in silico* binding energy between ML277 and KCNQ1 bound with PIP2 or KCNE3.

ML277 is an interesting and potentially important IKs activator: it can amplify IKs as long as the channel is not saturated with KCNE1, and it is selective for KCNQ1 but has little effects on the other KCNQ family channels. Because these features make ML277 an almost ideal IKs activator, there was a high degree of interest in how it works and where it binds. This high interest leads to a huge body of literature. Although there have been attempts to determine ML277's binding site in KCNQ1, using molecular docking and simulations, there is no 3D structure of KCNQ1 bound with ML277. This study is the first one reporting 3D structures of KCNQ1 in apo state and with ML277. Therefore, it is highly significant.

In addition to the significance, there are other important strengths in this article. The experiments were done with state-of-the-art technologies, with overall model resolution at 3.84 Å (xQ1-CaM) and 3.9 Å (xQ1-CaM-ML). Unequivocal identification of ML pose in the xQ1 structure was based on two independent apo xQ1 structures (the current one and the one published by MacKinnon), and two xQ1-CaM-ML structures. The extensive mutational studies support the importance of residues in the putative ML277 binding site based on the cryoEM data, and further reveal some unexpected features about ML277 (more below).

However, there are problems in some of the figures and related text. The authors need to clarify several points to make the article more balanced and more readable. These issues are listed below:

1. Fig. 3 and related text are meant to argue why ML277 is selective for Q1 based on the available cryoEM data, and how PIP2 may affect ML277 binding. Unfortunately they are confusing and not convincing: (a) The color scheme in panel A does not allow readers to clearly distinguish the 7 molecules embedded in the KCNQ channel. (b) What is the relevance of linopirdine, a Q1/Q3 inhibitor, in the context of KCNQ activators? (c) The related text is ineffective, fragmented, and not convincing. It is not clear how one side chain can wipe out the ML277 binding site or make ML277 binding unstable in non-Q1 channel. The whole argument is based on the authors' claim. Could you be more quantitative, e.g. calculating binding energy using molecular docking of ML277 to the corresponding sites in cryoEM Q2 and Q4 structures? A previous report showed that depleting membrane PIP2 (by strong depolarization

to activate coexpressed ci-VSP) enhances ML's activator effect on KCNQ1 (doi.org/10.1016/j.bpj.2014.10.059). This should be cited to support the model prediction.

2. The same concerns apply to Fig. 4, which is meant to explain why KCNE3 can destabilize ML277 binding to KCNQ1. It has been shown experimentally that progressive increase in KCNE3 (or KCNE1) decreases the ML277 activator effect (doi.org/10.1073/pnas.1300684110). There can be several contributing factors: e.g. KCNE blocks the entrance of ML277 to its binding pocket by steric hindrance at the membrane-cytoplasm interface. It is not clear whether side chains rotation alone can dislodge or prevent ML277, as is suggested here.

3. Fig. 5 presents extensive experimental results, but improvements are needed. (a) Arrange panel 'B' and panel 'C' side by side, so that the readers can see that mutations could separate the two ML277 effects on KCNQ1: current amplitude and voltage-dependence of activation. For example, ML277 induces a huge increase in the tail current amplitude of S338A and V255A, yet barely shifts the $V_{0.5}$ of activation of these two mutants. (b) The superimposed tail current traces currently shown in 'A' should be made into separate panels, each panel shows superimposed tail currents recorded from one channel in control and in ML277. This will clearly show that ML277 abolishes the hook in those tail currents that have hook under the control conditions. With this, current panel 'E' is not necessary and should be removed.

4. Fig. 6 is confusing. The purpose is to support the authors' claim that G272C has single channel behavior similar to S338A bound with ML277. The all-point histogram plots and Gaussian fits are shown in separate panels (B and D), and the single channel amplitudes do not look similar. Furthermore, there is no way for readers to judge whether the flicking kinetics is similar between the two. Overall, this figure does not serve useful purpose and should be removed.

5. Is there one and only ML277 binding site in KCNQ1, or not? ML277 is a small molecule that is expected to exhibit multiple conformations, making contacts with different sets of amino acid side chains depending on its location within the channel and its conformation. Such multiplicity in ML277 locations, conformation, and dynamic interactions with side chains in its surroundings was clearly seen in a previous molecular dynamics study. The fact that only one ML277 binding pose and a single binding pocket are identified in this study has a lot to do with how the final refined structures were achieved. More than 96% of the particles of xQ1-CaM-ML were removed (from 1,850,000 to 71,241) to reach the degree of resolution reported here. The 96% discarded particles may have other ML277 binding locations/poses that are legitimate. Furthermore, a live KCNQ1 channel goes through conformational changes during voltage-dependent gating, which can dramatically shift the ML277 binding site and likely the ML277 conformation. These real-world scenarios are not considered here but should not be forgotten. The authors initially did not rule out the possibility of other ML277 binding sites/poses (lines 173-175, Results). However, their tone changed dramatically in the 'Discussion' (line 462). This reviewer would advise the authors to clearly lay out the limitations of this state-of-the-art cryoEM structural determination, and present their findings in a more appropriate context.

6. The authors did not detect any major change in xQ1-CaM with 4 ML277 molecules bound within the channel, except a twist motion at the C-terminal coiled-coil region (between 1.4 and 2.9 Å) and rotation of some amino acid side chains at the binding site. This lack of detectable molecular motion after ML277 binds seems incompatible with the dramatic increase in current amplitude (signifying more ion conduction through the pore in PD, related to the 'removal of inactivation' mechanism of ML277), and

the negative shift in the voltage-dependence of activation (signifying changes in VSD, and/or linkage between VSD and PD). It should be pointed out that this is similar to the previous cryoEM report (ref 40), where Q4 channel activator or inhibitor binding did not induce any major changes in channel conformation. Is it possible that this phenomenon is related to how the cryoEM data are refined? If only a small % of particles consistent with a reference conformation are selected for refinement, this could lead to one channel conformation being 'identified' that is most prevalent but does not represent other more transient/less prevalent conformations relevant for channel function. Could the authors comment on this?

Reviewer #4 (Remarks to the Author):

KCNQ1 is responsible for the slow delayed rectifying K⁺ current (I_{ks}) in cardiac cells which recovers the cell from action potentials. Dysfunction of KCNQ1 can result in serious human diseases such as Long and Short QT syndromes. Therefore, developing drugs with high efficiency and specificity towards KCNQ1 is of great importance to cure related diseases. ML277 is a molecule that promisingly meets such requirements, exemplifying its importance. The manuscript by Willegems et al here reported the cryo-EM structures of xKCNQ1 with and without ML277 bound, uncovered potential binding site for ML277 on KCNQ1. The study also compared the binding site of ML277 with regulators from previously reported structures that bind preferentially to neuronal KCNQ channels and pointed out the differences. The overall quality of this study justifies its publication in Nature Communication. However, I have to say that presentation of the data in the manuscript is too lousy. I will agree to its publication only when the data is presented more clearly and logically.

Specific major concerns:

1. Line 189, 366 and Fig. 1E: About the “small displacement of the CTD” or “visible twist of the CTD”, it is most likely that such difference stems from the inherent flexibility of the CTD instead of the ligand binding. This point can be justified by the authors’ observation that even two different classes from same sample preparation have such a structural deviation of the CTD (line 191). Therefore, to avoid misleading, I would suggest removal of such descriptions and Fig 1E. The removal will not diminish the major finding in this study which is the binding details of ML277 on KCNQ1.
2. Fig. 3B: (1) The comparison is between ML277 and ML213. So why to portrait the molecule RTG here which makes the figure more crowded? (2) Val248 should be labeled in the figure and it should be labeled as Val248’ for clarity. For the same reason, Leu306, Phe325 should also be labeled as Leu306’, Phe325’.
3. Line 233-255: I don’t think it is necessary to include “ML277 versus PIP2” as an independent topic to discuss for that the two have little interactions. PIP2 is responsible for coupling the shift of S4 with inner

gate's opening while ML277 might function by promoting the conformational change in the TMD domain. They function independently, albeit having synergistic effect, and don't have to interact with each other.

4. Fig.3C: Related to point 3, I think the overlap of lipid tails and ML277 in different structures doesn't mean that ML277's binding given the flexibility of the lipid tails and the fact that lipid tail has no specific interaction with the transmembrane domain of the channel. It should be noted that the binding pocket of ML277 should be occupied by different kinds of lipid tails when the ligand is absent.

5. Fig.3D: Also related to point 3, (1) The described structural deviation between hKCNQ1-CAM-PIP2-KCNE3 and xKCNQ1-CAM-ML277 cannot be interpreted as the PIP2 binding-induced distortion of ML277 binding site. It makes more sense that this structural deviation is a result of the binding of ML277 to KCNQ1. I mean, when you determine the structure of hKCNQ1-CAM-PIP2 with adding ML277, it is possible that the binding site is quite similar with the one uncovered here in this study. Besides, I think this point is not a crucial one, thus I suggest moving the panel to extended data. (2) The comparison does not have to involve specific residues especially considering the not-very-high resolutions. Why not remove those residue labels to make the figure less crowded?

6. Fig.4C describes the difference between KCNQ1 and KCNQ4 to accommodate ML277. This point, I think, has been made clear in Fig3B, right? Why repeat it here?

7. Fig.4D describes different rotamers of Leu256 when binding with or without ML277. It is more reasonable to be moved to Fig.2.

8. Fig.4D: The view cannot clearly show the density difference of Leu256 side chain. It will be better if the helix is rotated a little bit clockwise when viewed from above.

9. Fig.5A,B: why do the authors prefer to use the tail currents recorded at -40mV instead of -120mV? To my knowledge, the tail currents recorded at -120 mV is larger and clearer than that recorded at -40mV. Either at -40 or -120 mV to record the tail currents, the channel undergoes the same inactivation process, right? Please explain this.

10. Fig.5C: It is better to also calculate and display $\Delta V_{1/2}$ values.

Minor points:

1. Line 72: should be "the subunit ratios of KCNE1:KCNQ1".

Reviewer #1 (Remarks to the Author):

The voltage-gated potassium channel KCNQ1 plays important physiological functions, and dysfunction thereof is linked to disease. Therefore, KCNQ1 is a highlighted pharmacological target and mechanistic insights into how KCNQ1 interact with small-molecule compounds is of great interest. In this study, Willegems and co-workers provide Cryo EM structures of KCNQ1 with and without the activator ML277, which allows the authors to propose a ML277 binding site and provide side-by-side comparison of structures without and with ML277 bound. The authors conclude that ML277 binds to a hydrophobic pocket formed by the S4-S5 linker and transmembrane segments 5 and 6, in which ML277 interacts with a large set of residues. In support of this, site-directed mutagenesis and electrophysiology experiments show that mutation of several of the proposed interacting residues alter the channel's response to ML277. Moreover, the authors take advantage of previously determined structures of KCNQ1/KCNE3, KCNQ2, and KCNQ4 to propose structural/mechanistic basis for ML277 selectivity. As such, the authors make several important contributions of relevance to the field. The manuscript is well written, and the extensive work described therein appears to be technically sound. However, given the authors expertise in ion channel biophysics and extensive previous experience with KCNQ1, the authors are encouraged to clarify/develop their mechanistic interpretations. This is particularly pertinent to their interpretation and conclusions related to parameters that determine the channel's response to ML277, but also applies to the putative role of PIP2 and KCNE subunits (see comments below).

We thank the reviewer for careful reading of our manuscript and recognition of its important contributions. In response to this reviewer and others we now include new sections in the Discussion where we address possible mechanisms by which ML277 exerts its actions at this location. These are entitled 'ML277 mechanism of action' and 'The C-helix twist'. Fundamentally, the discussion in these sections and the subsequent section 'ML277 binds to a region critical for inactivation', comes down to the presence of the binding pocket within the transmembrane domains at the locus of many key channel gating events: voltage-sensor pore coupling; pore gating; inactivation; and deactivation, all of which are channel kinetic properties modulated by ML277. More specific responses to questions are written below.

General comments:

1) As described in previous work and the present study, ML277 has several activating effects on KCNQ1, such as shifting $V_{1/2}$ of activation, increasing the overall tail current, and preventing channel inactivation. However, it is not evident how ML277 induces these different effects from a single binding site. Could the authors please discuss how they envision ML277 to induce these effects from the binding site identified in this study? In this context, please also comment on the finding that the structure with ML277 bound is in a closed conformation, despite ML277 being a channel activator. As pointed out by the authors themselves (row 184-189), the ML277 binding site might be distorted in other (open) conformations, the details of which the present structure does not provide insights into.

In response to request from a number of the reviewers we now include discussion of possible mechanisms of action of ML277 in specific sections entitled 'ML277 mechanism of action', 'The C-helix twist', and 'ML277 binds to a region critical for inactivation'. Due to length we have not reproduced those sections here and we refer the reviewer to the revised manuscript.

We have also now docked ML277 onto the open hKCNQ1-CaM-PIP2 structure (PDB-ID: 6V01) with KCNE3 removed, and find the binding site only changed in a minor way from that in the closed channel, and we calculate very similar binding energies. This result suggests that the drug can remain bound in both the open and closed pore conformations of the channel.

2) The authors provide an impressive amount of electrophysiology data to functionally assess the importance of KCNQ1 residues predicted to interact with ML277. The authors use the pharmacological

response of these mutants as a readout for preserved or altered ML277 binding (e.g. row 346-348). At the same time, the authors appreciate the importance of channel inactivation for ML277 effects, for instance as a possible mechanistic basis for lack of ML277 effects on KCNQ1/KCNE1 (which does not inactivate, row 448-454). This raises the question as to what extent altered response of proposed binding site mutants to ML277 is caused by impaired binding (validating the ML277 binding site) or could be explained by altered inactivation behaviour of mutants (not validating the ML277 binding site). Could the authors please clarify and motivate when they interpret impaired ML277 binding or altered intrinsic inactivation as the cause of smaller effects?

Altered intrinsic inactivation is clearly an important factor in the action of ML277 on binding site mutants, as the magnitude of the ML277 tail current response (increase in current) is well correlated with the pre-existing inactivation properties of each mutant (Fig. 7). Thus, S338A which shows augmented inactivation in control is most responsive to ML277 in terms of current increase, while G272C/T/L/V mutants which don't inactivate do not show any increase in current when exposed to ML277. There are, though, some anomalies. S338F retains 25% inactivation, but current is decreased in the presence of ML277. Ala mutations at L262 and L266 inactivate almost as much as WT in control, but currents are increased much less, while changes in the $V_{1/2}$ of activation are greater. F335A still inactivates 40%, but tail currents only increase 2x and the $V_{1/2}$ of activation actually depolarizes in the presence of ML277.

There are also other electrophysiological effects associated with the action of ML277 that we observe. These include slowing of the deactivation rate and changes in the $V_{1/2}$ of activation, and here the actions of ML277 on some mutants can be clearly separated from their inactivation characteristic, as seen in the new Fig. 6B if the decay rate of WT and S338A tails are compared. Although ML277 increases S338A tails 2x more than WT, the deactivation rate is slowed much less, and there is little effect on the $V_{1/2}$ of activation (Fig. 6D). In V255A, tails in control are very similar to WT (Fig. 7A, inset), and currents increase the same amount upon exposure to ML277, but there is no change in the $V_{1/2}$ of activation (Fig. 6C-E).

The data support the idea that individual mutations alter the ability of ML277 to interact with different parts of its binding site and thus disrupt different aspects of the electrophysiological actions of ML277, rather than the response to ML277 simply being determined by the prior inactivation property of the mutant. These ideas are now incorporated in the description of Fig. 6 in the Results section.

3) In relation to this, please comment on how the authors model aligns with the previously described effect of ML277 on KCNQ1/KCNE1 with less than four KCNE1 subunits in the complex. Is the effect possible due to one or several available ML277 binding sites not occupied by KCNE1 or because of the inactivation behaviour of unsaturated KCNQ1/KCNE1 complexes?

The reviewer raises an interesting point here, the role of accessory KCNE1 subunits in preventing the action of ML277. It has been shown that two KCNE1 (Yu et al. 2013), or even one (Eldstrom et al. 2021), reduces the effect of ML277 about 75%. At the single channel level ML277 significantly increases the open probability of KCNQ1 channels alone (by increasing open state stability and/or conductance), and the presence of KCNE1 subunits greatly reduces this action. Following on from the reviewer's comment, the implication is that if inactivation is reduced in unsaturated KCNQ1/KCNE1 complexes, then it is possible that the action of ML277 is correspondingly reduced, whether or not all available binding sites are occupied by ML277. We have looked at the inactivation of KCNQ1/KCNE1 in a 4:2 stoichiometry (see Fig. Rev1 below) and find that the tail hook is removed and therefore inactivation is prevented, making this likely an important factor in the reduction of ML277's effect in KCNQ1 complexes containing KCNE1.

Figure Rev1. IKs with 2:4 stoichiometry does not obviously inactivate. (A) Current traces for EQQ channels. Cells expressing EQQ were pulsed from -90 HP to potentials ranging from +60 down to +30 for 4 s and then to -120 mV for 0.9 s. (B) Close up look at the -120 mV tail portion of EQQ traces.

4) The phospholipid PIP2 is required for electromechanical coupling in KCNQ1. The authors describe that the ML277 binding site is anticipated to be distorted upon PIP2 binding (row 250-255). Please discuss the anticipated functional consequence of PIP2 binding in the context of ML277's ability to bind to its site and act as a channel activator.

The present cryo-EM experiments do not directly address the effect of PIP2 binding to the complex, so in order to answer the reviewer's question, we carried out docking of ML277 to the published open pore hKCNQ1-CaM-PIP2 structure (PDB-ID: 6V01) after removal of KCNE3 (Fig. 3) and MD simulations of the ML277 and PIP2 poses during 200 ns runs (Supplementary Fig.9). The results indicate that PIP2 tail flexibility allows ML277 access to its binding pocket in the open channel, which is not greatly altered from that in our xKNCQ1-CaM structure.

5) The slightly twisted C-terminal coiled coil domain in the ML277 bound structure compared to unbound structures is highlighted throughout the results section. However, the putative functional relevance of this twist is not commented upon. Please discuss.

In both of the data sets of resolved structures of ML277 bound to KCNQ1 there was a clear twist to the C-terminal coiled coil domain, which was absent in both our structure of KCNQ1 alone and that published before (PDB-ID: /5VMS), as shown better in a revised Supplementary Fig. 7. We now discuss this twist in a new Discussion section. Alignment of the activated and open inner gate structure of human KCNQ1 (paired with KCNE3, PDB-ID: 6v01; Sun and MacKinnon, 2020) with our apo and ML277 bound structures shows that in order to account for the C-terminal rearrangements necessary for channel opening, this C-helix must move up towards the membrane and the upper half of the helix transitions to a poorly resolved flexible structure. How binding of ML277 in the transmembrane segment leads to this twist is not obvious, but it is possible that there are subtle conformational changes in the area between the ML277 site and the coiled-coil, too small to reliably detect at the current cryo-EM resolution. This pre-twist may prime the channel to open more readily. Future electrophysiological experiments would be required to investigate this possibility.

Specific comments:

The authors comment upon “additional density, likely representing unmodeled detergent or lipid molecules” in the Figure 1 legend. Is there any chance these densities represent ML277 in alternative sites?

In response to the reviewer’s request we have further examined the unmodeled densities from the xKCNQ1-CaM and both xKCNQ1-CaM-ML277 data set analyses. The results are shown in a revised Supplementary Fig. 8. Panel E, overlays the unmodeled densities from the apo and drug-bound structures. We found no xKCNQ1-CaM-ML277 densities (red in Supplementary Fig. 8E) that were not reproduced in the apo xKCNQ1-CaM structure (white in Supplementary Fig. 8E). The reader is directed to Supplementary Fig.8 from the Fig.1 legend.

We now add a comment in the second paragraph of the Discussion, stating that we cannot exclude alternative binding locations for ML277 during activation and other gating conformations which our structure and other existing structures do not address.

Line 104: “neuronal Kv7 channels” could be “neuronal KCNQ channels” to consistently use KCNQ nomenclature.

Changed

Line 147: Please include numerical values (mean +/- sem) for ML277 effects on the truncated xKCNQ1, to allow for quantitative comparison to the effect on human KCNQ1.

We have performed additional experiments on xKCNQ1 and now show these in a revised Supplementary Fig.1, along with mean +/- sem. The data show similar ML277 sensitivity between the truncated xKCNQ1 used in cryo-EM experiments and human full-length KCNQ1 expressed in mammalian cells.

Line 182: “kCNQ1” should be “KCNQ1”.

Corrected

Line 196: “...hindered sterically...”. I find this phrasing hard to understand.

We have edited this phrase to make it clearer to the reader.

Line 228-229: Could the role of the side chain bulkiness for channel selectivity of listed residues possibly be supported by mutagenesis experiments?

We have supported the role of side chain bulk in channel isoform and drug selectivity using MD simulations (new Fig. 4 and Fig. 5). The results and binding energies clearly support the underlying basis for ML277 selectivity for KCNQ1 and ML213 selectivity for KCNQ2 and 4.

Line 283: Please clarify that this effect is for 1 μ M of ML277.

This is for 1 μ M ML277, now written into the text.

Line 293-294: This sentence is hard to understand.

Sentence has been rewritten to improve clarity

Figure 1 legend: Please explain what HA-HD denotes.

This has now been clarified as helices A-C of the KCNQ1 C-terminus in the legend and the associated text.

Figure 5C-D: Why is not sem included?

The sem was omitted. It has now been included in panels C, D, and E of Fig.6.

Supplementary Figure 1D and 9: Please explain how normalization was done.

Normalization was performed to the peak tail current value during the protocol for each cell to reduce the effect of current amplitude variations between cells. The mean data before normalization were shown in

Supplementary Fig. 11B for those interested. We now explain this measurement in the Methods section and legend to Supplementary Fig. 1.

Supplementary Figure 1E: is this a single representative recording? Otherwise, please include sem.

The single example of the G-V relationship in Supplementary Fig.1E has now been replaced by the mean of 6 cells in Supplementary Fig.1B,

Supplementary Figure 2A: “xKNQ1-CaM” should be “xKCNQ1-CaM”.

Corrected. Thank you for noticing this error.

Supplementary Information: The title is not consistent with the title of the main work.

Corrected

Reviewer #2 (Remarks to the Author):

In the manuscript, Willegems and colleagues describe the binding site of KCNQ1 activator ML277 utilizing cryo-electron microscopy and electrophysiological techniques. The study offers novel and accurate structural insights into how the KCNQ1 activator binds to the channel and how corresponding gating modifications may occur. Comparison of cryo-EM structures obtained with and without ML277 revealed a binding conformation of a drug and explains the isoform specificity of its action. Authors show that the binding pocket of the channel is formed by several residues of S4-S5 linker, S5 and S6 pore helices of one subunit as well as by some residues in S5 and S6 segments of the neighboring subunit. The structures likely represent a configuration of the channel with activated voltage-sensor and closed pore, which in line with what has been reported for KCNQ1 channels without obligatory ligand PIP2. The absence of PIP2 in the structures makes it difficult for authors to interpret the structural data in context of the precise mechanism of ML277 action. However, authors performed a detailed mutational analysis of the key residues coordinating ML277 revealed from structural data and could show that these residues also play an important role in the drug-induced gating modifications for fully functional channels that likely have PIP2 molecules bound to their structure. The study has important implications for understanding the mechanism of the ML277 action on KCNQ1 channels, which is an imperative step towards the design of a more potent and highly specific analog(s) of the drug for potential treatments of KCNQ1-related pathologies. I only have a few minor comments that I hope will help authors to improve the manuscript:

We thank the reviewer for their careful reading of our manuscript and constructive comments.

1) A remarkable modification of wild type homomeric KCNQ1 channel by ML277 is that the drug renders the channel constitutively open. The fraction of constitutively open channels is quite large already at 1 μ M concentration of the drug as electrophysiological experiments indicate. Similar results were also reported earlier by authors. Represented structures, nevertheless, indicate that the inner gate of the channel is closed. This could be due to the fact that PIP2 is missing from the KCNQ1-CAM-ML277 complex. A concise and comprehensive discussion about the putative mechanism of how ML277 may render the channel constitutively open in the context of the presented structural data would be very interesting.

We have now carried out docking and MD simulations of ML277 bound to the hKCNQ1-CaM-PIP2 structure (Revised Fig. 3 and Supplementary Fig. 9) to understand the binding site for ML277 in the open pore structure and in the presence of PIP2. The results indicate only minor changes to the pocket for ML277 with binding energies of \sim 50 kcal/mol in xKCNQ1-CaM and the open state with PIP2 in hKCNQ1-CaM-PIP2. So, clearly, ML277 can bind in a similar pocket in the open and closed channel, with and without PIP2. There are a number of possibilities how ML277 at this site regulates gating. As stated above in the response to Reviewer 1, comment #1, new sections are included in the Discussion to consider mechanisms by which ML277 may exert its three-fold electrophysiological actions. It should be noted that the binding

pocket lies in a position where subtle changes to activation energetics and/or pore opening and closing kinetics could well account for the electrophysiological actions observed.

2) Related to the point 1: the extent of ML277-induced current potentiation in S338A mutant is roughly twice as large compared to wild type. However, ML277 does not seem to have a large constitutively active component. The same is true for V255A mutant – increase of tail current is comparable with wild type but no large constitutively open fraction is observed. What is the explanation for these discrepancies?

These mutants are individual binding site mutations and most of the effect of ML277 in these two cases is due to an increase in current size (perhaps related to removal of fast inactivation). There is no increase in constitutive current because in these mutants the channels still deactivate almost fully during the 10 s interpulse interval. This can be clearly seen in the new Fig. 6B if the decay rate of WT and S338A tails are compared. Thus, there is little effect on the $V_{1/2}$ of activation as well for V255A or S338A. The data support the idea that disruption of all aspects of ML277 binding are required to prevent all its electrophysiological actions, and neither V255A, nor S338A, achieve that.

3) In the manuscript (lines 238-240) authors write “... ML277 as a KCNQ1 activator does not mimic the action of PIP2 on the channel conformation, at least under the conditions used for this cryo-EM study”. For readers it is unclear why authors expect ML277 to mimic the action of PIP2 molecules. To my knowledge, no functional data are so far reported that show an effect of the drug that resembles PIP2-like action on KCNQ1 channels. In addition, the reported amino acids that coordinate PIP2 in KCNQ1, with exception of W248, are different from those that coordinate ML277. This section could be improved so that the framework of the arguments become clearer to the readers.

Here we just wanted to point out that ML277 and PIP2 are both channel activators, and not that ML277 had been proposed to mimic PIP2 action. However, PIP2 is known to cause large changes in channel conformation (Sun and MacKinnon, 2020), but ML277 does not. We have rephrased this sentence to improve clarity.

4) The normalized peak tail current shown in the Supplementary (Fig. 9) for L251A mutant does not correspond to the peak currents shown in the right panel. This is evident when the current levels corresponding to potentials more negative than -20 mV are considered.

There was an erroneous graph (red points) placed in panel A due to a normalization error. The correct graph was shown in panel B, and they clearly do not match. This issue has now been fixed in Supplementary Fig. 11, and we thank the reviewer for picking up the error.

Reviewer #3 (Remarks to the Author):

This study by Willegems et al reports cryoEM structures of *Xenopus* KCNQ1 associated with Ca-calmodulin in the absence (xQ1-CaM) or presence of ML277 (xQ1-CaM-ML277) (Figs. 1 and 2). The authors tested whether mutations in the putative ML277 binding site (made in the human KCNQ1 ortholog) could alter ML277's effects on this channel (Fig. 5A-C, Supplementary Fig. 9). They used these functional data, as well as some single channel data (Fig. 6), to support the statement that ML277 increased KCNQ1 current amplitude by removing KCNQ1 inactivation. Finally, the authors extensively discussed why ML277 is relatively specific toward KCNQ1 vs the other members of the KCNQ family, and how PIP2 and KCNE3 can affect ML277 binding to KCNQ1 (Figs. 3 and 4). This latter part is solely based on alignments of their structure(s) with published cryoEM structures, instead of any quantitative analysis, such as calculation of *in silico* binding energy between ML277 and KCNQ1 bound with PIP2 or KCNE3. ML277 is an interesting and potentially important IKs activator: it can amplify IKs as long as the channel is not saturated with KCNE1, and it is selective for KCNQ1 but has little effects on the other KCNQ family channels. Because these features make ML277 an almost ideal IKs activator, there was a high degree of interest in how it works and where it binds. This high interest leads to a huge body of literature. Although

there have been attempts to determine ML277's binding site in KCNQ1, using molecular docking and simulations, there is no 3D structure of KCNQ1 bound with ML277. This study is the first one reporting 3D structures of KCNQ1 in apo state and with ML277. Therefore, it is highly significant.

In addition to the significance, there are other important strengths in this article. The experiments were done with state-of-the-art technologies, with overall model resolution at 3.84 Å (xQ1-CaM) and 3.9 Å (xQ1-CaM-ML). Unequivocal identification of ML pose in the xQ1 structure was based on two independent apo xQ1 structures (the current one and the one published by MacKinnon), and two xQ1-CaM-ML structures. The extensive mutational studies support the importance of residues in the putative ML277 binding site based on the cryoEM data, and further reveal some unexpected features about ML277 (more below).

However, there are problems in some of the figures and related text. The authors need to clarify several points to make the article more balanced and more readable. These issues are listed below:

We thank the reviewer for their supportive comments concerning the strengths of our article, the first reporting of KCNQ1 in an apo- and drug-bound state, the cryo-EM resolution, and extensive mutational data in support of the structural determination. We acknowledge the lack of in-silico binding energy calculations, and have remedied this by adding docking studies of ML277 into the open KCNQ1-CaM-PIP2 structure (Fig. 3), as well as into hKCNQ2 and hKCNQ4 (Fig. 4). We have carried out additional simulations of PIP2 configurations in the presence of ML277 to understand spatial clashes between the two and the range of potential poses is shown in Supplementary Fig. 9.

1. Fig. 3 and related text are meant to argue why ML277 is selective for Q1 based on the available cryoEM data, and how PIP2 may affect ML277 binding. Unfortunately they are confusing and not convincing: (a) The color scheme in panel A does not allow readers to clearly distinguish the 7 molecules embedded in the KCNQ channel. (b) What is the relevance of linopirdine, a Q1/Q3 inhibitor, in the context of KCNQ activators? (c) The related text is ineffective, fragmented, and not convincing. It is not clear how one side chain can wipe out the ML277 binding site or make ML277 binding unstable in non-Q1 channel. The whole argument is based on the authors' claim. Could you be more quantitative, e.g. calculating binding energy using molecular docking of ML277 to the corresponding sites in cryoEM Q2 and Q4 structures? A previous report showed that depleting membrane PIP2 (by strong depolarization to activate coexpressed ci-VSP) enhances ML's activator effect on KCNQ1 (doi.org/10.1016/j.bpj.2014.10.059). This should be cited to support the model prediction.

We agree with the reviewer and have now revised this figure and its associated text in accordance with all the reviewer's wishes. We have removed linopirdine as requested, enlarged the figure panel and improved the colors of other activators. We have separated our description of the actions of ML277 into: 1) the open pore structure of KCNQ1, PIP2, and the inhibitory role of KCNE3 (Fig. 3); 2) the selectivity of ML277 for KCNQ1 over KCNQ2 and KCNQ4 using docking simulations to other published structures and quantitated the decreased binding score of the drug to these channels compared with Q1 (Fig. 4); and 3) made a comparison of ML213 and ML277 binding to KCNQ1 and KCNQ4 (Fig. 5). We now cite Xu *et al* 2015 in the Results section text to in Fig. 3, in description relevant to PIP2.

2. The same concerns apply to Fig. 4, which is meant to explain why KCNE3 can destabilize ML277 binding to KCNQ1. It has been shown experimentally that progressive increase in KCNE3 (or KCNE1) decreases the ML277 activator effect (doi.org/10.1073/pnas.1300684110). There can be several contributing factors: e.g. KCNE blocks the entrance of ML277 to its binding pocket by steric hindrance at the membrane-cytoplasm interface. It is not clear whether side chains rotation alone can dislodge or prevent ML277, as is suggested here.

We have further examined the positions of residues in S5 (L256 and F270) in the hKCNQ1-CaM-PIP2 structure in the absence and presence KCNE3, compared with their positions in our xKCNQ1-CaM-ML277 structure (Fig. 3). MD simulations of ML277 binding to 6V01 in the absence of KCNE3 showed these side chains to adopt positions to accommodate the 4-methoxyphenyl group of ML277, close to their orientations in our xKCNQ1-CaM-ML277 structure (Fig. 3B). However, in the presence of KCNE3, the forced positions

of side chains of L266 and F260 in the presence of KCNE3 clash severely with ML277. In the xKCNQ1-CaM-ML277 structure the χ_1 angle (rotatable bond between $C\alpha$ and $C\beta$) of Leu256 is 173° , and in the hKCNQ1-CaM-PIP2-KCNE3 structure is -72° . In the presence of KCNE3, if the χ_1 angle is more negative than -80° it clashes with F270, and if it is higher than 50° it clashes with T71, which demonstrates that the χ_1 angle in the presence of KCNE3 cannot be 173° as in the case of xKCNQ1-CaM-ML277. This improved analysis is presented in the text to Fig. 3.

3. Fig. 5 presents extensive experimental results, but improvements are needed. (a) Arrange panel 'B' and panel 'C' side by side, so that the readers can see that mutations could separate the two ML277 effects on KCNQ1: current amplitude and voltage-dependence of activation. For example, ML277 induces a huge increase in the tail current amplitude of S338A and V255A, yet barely shifts the $V_{0.5}$ of activation of these two mutants. (b) The superimposed tail current traces currently shown in 'A' should be made into separate panels, each panel shows superimposed tail currents recorded from one channel in control and in ML277. This will clearly show that ML277 abolishes the hook in those tail currents that have hook under the control conditions. With this, current panel 'E' is not necessary and should be removed.

We have now re-arranged this figure as the new Fig.6. The electrophysiological summary effects of ML277 can be now seen side-by-side in panels C, D, and E. The tail currents at -120 mV are shown separated in Fig.6B, as requested, each with their respective control. The old Panel E has been removed from this Figure.

4. Fig. 6 is confusing. The purpose is to support the authors' claim that G272C has single channel behavior similar to S338A bound with ML277. The all-point histogram plots and Gaussian fits are shown in separate panels (B and D), and the single channel amplitudes do not look similar. Furthermore, there is no way for readers to judge whether the flicking kinetics is similar between the two. Overall, this figure does not serve useful purpose and should be removed.

Fig.6 has been repurposed as a new Fig. 7 to show the correlation of inactivation with ML277 effect in binding pocket mutants. The single channel data have been revised and are presented differently. There is a strong correlation between the degree of inactivation in control and the magnitude of the ML277 tail current increase (Fig. 7A). We still feel that showing the marked change in single channel behaviour between single channels of WT and S338A exposed to ML277, compared with G272C in control, is valuable. The data provide supportive single channel evidence to explain why S338A responds more strongly than WT to ML277, and why G272C does not respond at all.

5. Is there one and only ML277 binding site in KCNQ1, or not? ML277 is a small molecule that is expected to exhibit multiple conformations, making contacts with different sets of amino acid side chains depending on its location within the channel and its conformation. Such multiplicity in ML277 locations, conformation, and dynamic interactions with side chains in its surroundings was clearly seen in a previous molecular dynamics study. The fact that only one ML277 binding pose and a single binding pocket are identified in this study has a lot to do with how the final refined structures were achieved. More than 96% of the particles of xQ1-CaM-ML were removed (from 1,850,000 to 71,241) to reach the degree of resolution reported here. The 96% discarded particles may have other ML277 binding locations/poses that are legitimate. Furthermore, a live KCNQ1 channel goes through conformational changes during voltage-dependent gating, which can dramatically shift the ML277 binding site and likely the ML277 conformation. These real-world scenarios are not considered here but should not be forgotten. The authors initially did not rule out the possibility of other ML277 binding sites/poses (lines 173-175, Results). However, their tone changed dramatically in the 'Discussion' (line 462). This reviewer would advise the authors to clearly lay out the limitations of this state-of-the-art cryoEM structural determination, and present their findings in a more appropriate context.

In our strategy to process the cryo-EM data, we used a low threshold, picking as many particles as possible, which includes a lot of 'noise'. We then rejected particles that clearly didn't represent xKCNQ1 channels at the 2D classification stage. Such a strategy ensures that we don't miss out on useful particles that can

contribute to the final models. We now show the 2D classification results in Supplementary Fig. 6, indicating that all rejected 2D classes do not represent discernible xKCNQ1 particles, but rather represent contaminants (e.g. micelles, ice). To further ensure there are no hidden xKCNQ1 particles in this rejected stack, we have now performed further 2D classification of this rejected stack into 50 classes to improve the signal-to-noise of the class averages. This again does not show any discernible xKCNQ1 particles.

We also looked at unmodeled densities in the independent xKCNQ1-CaM (n=1) and xKCNQ1-CaM-ML277 (n=2) density datasets (Supplementary Fig. 8). We did not find any unique densities in the xKCNQ1-CaM-ML277 maps that could fit ML277. We can conclude from our experiments that we have only detected a single ML277 binding pocket in xKCNQ1-CaM.

We completely agree, though that this does not exclude the presence of other ML277 binding poses in images we did not capture, or during gating itself, as suggested above. We take some comfort that the binding pocket that we have identified does not change much from the closed xKCNQ1-CaM structure to the open hKCNQ1-CaM-PIP2 structure as shown by the very similar binding energies that we calculate (Fig. 3). Nevertheless, many gating conformations are traversed during opening and closing pathways about which we have no information. We have included these ideas in a section at the start of the Discussion to highlight to the reader our awareness of these important issues.

6. The authors did not detect any major change in xQ1-CaM with 4 ML277 molecules bound within the channel, except a twist motion at the C-terminal coiled-coil region (between 1.4 and 2.9 Å) and rotation of some amino acid side chains at the binding site. This lack of detectable molecular motion after ML277 binds seems incompatible with the dramatic increase in current amplitude (signifying more ion conduction through the pore in PD, related to the 'removal of inactivation' mechanism of ML277), and the negative shift in the voltage-dependence of activation (signifying changes in VSD, and/or linkage between VSD and PD). It should be pointed out that this is similar to the previous cryoEM report (ref 40), where Q4 channel activator or inhibitor binding did not induce any major changes in channel conformation. Is it possible that this phenomenon is related to how the cryoEM data are refined? If only a small % of particles consistent with a reference conformation are selected for refinement, this could lead to one channel conformation being 'identified' that is most prevalent but does not represent other more transient/less prevalent conformations relevant for channel function. Could the authors comment on this?

As mentioned in point 5, it is unlikely that we are missing out on a significant number of KCNQ1 particles, as the rejected particles seem to represent contaminants. We have also performed 3D classification, yielding two distinct classes with a different 'twist' of the C-terminal coiled-coil (Supplemental Fig. 7). But no other 3D classes with significant changes in, for example, the PD or VSD, could be picked up. We cannot rule out that there are some particles with a different conformation of the TM region, which end up in Class 1 or Class 2 of the xKCNQ1-CaM-ML277, but clearly these would only represent a tiny fraction, as they would otherwise have led to much lower resolution of the TM region. Thus, the predominant form of xKCNQ1-CaM-ML277 is in a conformation that is very similar to xKCNQ1-CaM.

We also now point out that our result is similar to the recent cryo-EM report for ML213, and in revised sections of the Discussion we have provided potential explanations why such a binding site at a key location for channel gating and pore opening may be enough to cause the observed electrophysiological effects. Nevertheless, we absolutely agree that it is important to express caution in presenting our results and we added a number of caveats in the first two pages of the Discussion.

Reviewer #4 (Remarks to the Author):

KCNQ1 is responsible for the slow delayed rectifying K⁺ current (I_{ks}) in cardiac cells which recovers the cell from action potentials. Dysfunction of KCNQ1 can result in serious human diseases such as Long and Short QT syndromes. Therefore, developing drugs with high efficiency and specificity towards KCNQ1 is of great importance to cure related diseases. ML277 is a molecule that promisingly meets such requirements, exemplifying its importance. The manuscript by Willegems et al here reported the cryo-EM structures of xKCNQ1 with and without ML277 bound, uncovered potential binding site for ML277 on

KCNQ1. The study also compared the binding site of ML277 with regulators from previously reported structures that bind preferentially to neuronal KCNQ channels and pointed out the differences. The overall quality of this study justifies its publication in Nature Communication. However, I have to say that presentation of the data in the manuscript is too lousy.

I will agree to its publication only when the data is presented more clearly and logically.

We appreciate the reviewer's comments regarding the overall quality of our study and in our revisions we have improved the organization of the figures and text and quantified our models and interpretation with additional experiments and simulations.

Specific major concerns:

1. Line 189, 366 and Fig. 1E: About the “small displacement of the CTD” or “visible twist of the CTD”, it is most likely that such difference stems from the inherent flexibility of the CTD instead of the ligand binding. This point can be justified by the authors' observation that even two different classes from same sample preparation have such a structural deviation of the CTD (line 191). Therefore, to avoid misleading, I would suggest removal of such descriptions and Fig. 1E. The removal will not diminish the major finding in this study which is the binding details of ML277 on KCNQ1.

It is possible that the reviewer has misunderstood our data here, as the twist of the CTD was noted in two sets of data from different sample preparations collected several months apart, and indeed using different collector devices (see revised Supplementary Fig. 7). The reviewer is correct that we obtained two different classes for the largest ML277 dataset showing different degrees of ‘twist’, indicating a certain degree of flexibility, but in both classes, there is still a clear twist relative to any xKCNQ1 structures without ML277. For example, the twist was not found in our KCNQ1 preparations without ML277, nor in the previously published structure (PDB-ID: 5VMS). Given the different datasets by ourselves and others, and the different classes, we have 3 maps for ML277-bound structures showing a twist, and 2 maps without ML277 that do not show a twist. So, the CTD twist cannot be attributed to specific sample and we have no doubt that it is a real effect of ML277 binding. It is important, therefore, that we retain these data in the present manuscript. The new Supplementary Fig. 7 now presents a detailed comparison of the experimental density for the coiled-coil region in all xKCNQ1 structures. As well, a more detailed description of the role of the C-terminal twist was requested by reviewer #1, and so we have kept Fig. 1E and speculate further on the role of the twisted CTD in the Discussion as requested, with a specific section entitled ‘C-helix twist’. We do acknowledge that there remains a certain amount of speculation around the role of the twist, but strongly prefer to keep this in the manuscript, as it may spur further functional investigations by the KCNQ field.

2. Fig. 3B: (1) The comparison is between ML277 and ML213. So why to portrait the molecule RTG here which makes the figure more crowded? (2) Val248 should be labeled in the figure and it should be labeled as Val248' for clarity. For the same reason, Leu306, Phe325 should also be labeled as Leu306', Phe325'.

We have separated the old Figs. 3 and 4 into new Figs. 3, 4, and 5, to improve clarity and add the new MD simulation data. We agree with the reviewer that Fig. 3B was crowded and have removed RTG from Fig. 4A. Apostrophes have been added where appropriate.

3. Line 233-255: I don't think it is necessary to include “ML277 versus PIP2” as an independent topic to discuss for that the two have little interactions. PIP2 is responsible for coupling the shift of S4 with inner gate's opening while ML277 might function by promoting the conformational change in the TMD domain. They function independently, albeit having synergistic effect, and don't have to interact with each other.

In order to understand the potential binding pocket of ML277 in the open channel we have docked ML277 to the published structure of hKCNQ1-CaM-PIP2 with KCNE3 removed (Fig. 3A, 3C), and carried out MD simulations for ML277 docking in this structure (Fig. 3B, 3D, Supplementary Fig. 9). We find that the PIP2 tails do interact with ML277 and feel that these simulations and docked structures inform the role of ML277 in the open channel.

4. Fig.3C: Related to point 3, I think the overlap of lipid tails and ML277 in different structures doesn't mean that ML277's binding given the flexibility of the lipid tails and the fact that lipid tail has no specific interaction with the transmembrane domain of the channel. It should be noted that the binding pocket of ML277 should be occupied by different kinds of lipid tails when the ligand is absent.

The reviewer is correct here and we agree. See Fig. 3, Supplementary Fig. 9 and response to point 3, above.

5. Fig.3D: Also related to point 3, (1) The described structural deviation between hKCNQ1-CAM-PIP2-KCNE3 and xKCNQ1-CAM-ML277 cannot be interpreted as the PIP2 binding-induced distortion of ML277 binding site. It makes more sense that this structural deviation is a result of the binding of ML277 to KCNQ1. I mean, when you determine the structure of hKCNQ1-CAM-PIP2 with adding ML277, it is possible that the binding site is quite similar with the one uncovered here in this study. Besides, I think this point is not a crucial one, thus I suggest moving the panel to Supplementary. (2) The comparison does not have to involve specific residues especially considering the not-very-high resolutions. Why not remove those residue labels to make the figure less crowded?

We agree with the reviewer's comments and have revised Fig. 3 to improve the content and impact. We believe that it now gives much more information to the reader. The full response is shown above to point 3.

6. Fig.4C describes the difference between KCNQ1 and KCNQ4 to accommodate ML277. This point, I think, has been made clear in Fig3B, right? Why repeat it here?

In the reorganization of figures, Fig. 4 compares the binding pocket for ML277 in xKCNQ1-CaM with potential binding pockets for ML277 in hKCNQ2 and hKCNQ4 (Fig. 4 A-D). MD simulations were carried out to obtain binding energies (which were greatly diminished) and poses for ML277 bound to hKCNQ2 and hKCNQ4 (Fig. 4E, F). We hope that this reorganization and extra modeling has improved the clarity and information contained in the comparison.

7. Fig.4D describes different rotamers of Leu256 when binding with or without ML277. It is more reasonable to be moved to Fig.2.

This was done in the reorganization of new Figs. 3-5 and is now presented as Fig. 2E.

8. Fig.4D: The view cannot clearly show the density difference of Leu256 side chain. It will be better if the helix is rotated a little bit clockwise when viewed from above.

This panel has been moved to Fig. 2E and the view has been rotated.

9. Fig.5A,B: why do the authors prefer to use the tail currents recorded at -40mV instead of -120mV? To my knowledge, the tail currents recorded at -120 mV is larger and clearer than that recorded at -40mV. Either at -40 or -120 mV to record the tail currents, the channel undergoes the same inactivation process, right? Please explain this.

There is little theoretical difference in recording tail currents at -40 or -120 mV as the instantaneous current is used to record the gating state for the prior voltage steps. With an equilibrium potential for K^+ of about -85 mV, tail currents at -40 mV should actually be a bit larger than at -120 mV. It is true that tsa201 cells do not like to spend long periods at -120 mV, so we tend to limit time spent there. We did show tails at -120 mV in Supplementary Fig. 11, and we now show more tails at -120 mV as asked for by Reviewer 2 and shown in Fig. 6B and 7A.

10. Fig.5C: It is better to also calculate and display $\Delta V_{1/2}$ values.

These values are already presented in Supplementary Table 3 and full activation curves in Supplementary Fig. 11A and 11B. We have decided to keep them there as the essential findings are represented in Fig. 6C-E and the data in Supplementary Fig. 11 remain a reference for those who wish to study the detailed electrophysiological data.

Minor points:

1. Line 72: should be “the subunit ratios of KCNE1:KCNQ1”.

This has been changed

REVIEWERS' COMMENTS

Reviewer #1 (Remarks to the Author):

The authors have done a good job in revising the manuscript and most of my previous concerns have been addressed. The authors now include extended discussions about the mechanisms underlying altered ML277 sensitivity of KCNQ1 mutations and putative mechanistic implications of the C-terminal twist and KCNE3. However, in the revised sections, I have some additional suggestions of clarifications to make:

1. I like that the authors have employed MD simulations on a version of the open KCNQ1 structure to further explore putative ML277 interaction with different conformational states and the impact of PIP2. However, some aspects of how the MD simulations were done are poorly described in the main text and SI methods. Please add necessary details such as: “its binding site refinement”, what is this?; “Optimized using ICM-Pro”, what is this; “the tetramerization domain was trimmed”, please indicate from which residue subsequent protein structure was omitted.
2. Similarly, for docking, it is hard to understand from the main text and figure legends when docking was made or when structures were just superimposed. Moreover, it is not clear to me whether docking with flexible side chains was always used (e.g., also when exploring the chi angle (row 250-256). Please make clear in each figure legend when data represent resolved structures with ML277 binding, docking of ML277 to structures to predict putative binding, or just superposition of structures to indicate putative differences in binding sites.
3. Row 244-264: The revised text on how KCNE3 may render the KCNQ1 channel insensitive to ML277 is improved. However, given the constitutively open phenotype of KCNQ1/KCNE3, it is still not clear to me whether one could expect ML277 effects on KCNQ1/KCNE3 (even if specific side chains were in a favourable rotation). Please comment.
4. Row 437-439: “We take some comfort that the binding pocket that we have identified does not change much from the closed... to the open... structure”. Could the authors comment on how they take comfort in this? Given the ML277 effects on KCNQ1, wouldn't it make more sense if the binding energy was greater in the open state?
5. Row 195 and 402: The authors refer to the “native structure”. Can they please clarify this? Is this the unbound apo structure?

6. Row 283: “ICM”, please define abbreviation.

7. Row 327: “reflects changes in the kinetics of VSD activation”, please clarify how the deactivation kinetics mainly reflects VSD activation kinetics.

8. Row 507: “electromechanical coupling (EMC)”, this abbreviation was already stated on row 502.

9. Row 534: “activated state”, do the authors mean open state?

10. It is great that the authors use a three-letter indication for xKCNQ1 residues and one-letter indication for hKCNQ1 residues. However, in the Discussion, there seems to be some mix-up (e.g., row 480 and 485). Please double-check if the correct indication is used throughout the Discussion.

11. Figure 5 legend title: “regulators to KCNQ1”. Perhaps change to KCNQ as not all regulators act on KCNQ1.

12. Supplementary figure 9: “Note that sidechain orientation now matches that seen in...”. Is the sidechain re-orientation caused by KCNE3 removal and subsequent relaxation of the model, or by relaxation per se? In other words, is the same re-orientation seen when running MD in the presence of KCNE3?

Reviewer #2 (Remarks to the Author):

The manuscript has been greatly improved and the revision was met with addition of significant amounts of new data. The updated Figures 2, 3, 4 are more illustrative. In this revised manuscript, the authors address previous concerns from me and other reviewers related to the binding pose of the ML 277 in the open channel, when the PIP2 molecule is also bound, as well as to the corresponding mechanism of action. These issues have been satisfactorily addressed by docking experiments and molecular dynamics simulations on open channel (PDF ID 6v01). The “ML 277 mechanism of action” section in the Discussion now reasonably explains the issues I raised related to ML277 action rendering

the channel constitutively open. Discrepancies of drug action on different mutants, including V255A and S338A, are discussed in the Discussion. I found only one error described below.

Page 10, line 223: Xua et al. 2014. The publication year should be changed to 2015.

Reviewer #4 (Remarks to the Author):

The authors have appropriately responded to my questions. No further revision is needed from me.

Reviewer #5 (Remarks to the Author):

It is a great article with a discovery of the binding model of this potassium channel activator ML277.

Some comments on the binding energy calculations using the obtained structural model and MMPBSA.

I understand that the numbers are beyond the scope of this manuscript but binding free energies around -50 kcal/mole are clearly grossly exaggerated for a small molecule binder such as ML277 since it translates into insanely strong binding constant (Kd orders of magnitude below femtomolar). I am not sure anything can be done about it however by the authors.

The docking of ML277 and some other ligands to the KCNQ1 model and its homologs (Q2,Q4) are very useful to understand the specificity and alternative binding pockets. It was great to see that ICM docking predicted binding mode close to the experimentally determined one.

I suggest some language changes:

l. 183: "which strongly supports the posited location for ML277" , I assume it is a typo, may be the authors meant "proposed"

line 283,284 : I suggest to say, "less favorable" instead of "lower". Term "lower" implies more favorable binding and -31 lower than -16.6, not higher since these are negative values. Using "less favorable" is one way to avoid going into all those details yet keep the meaning clear.

The ICM docking scores for KCNQ2 and KCNQ4 were -16.6 and -9.42, respectively, which is much less favorable than for xKCNQ1-ML277 (-31.1).

line 285. I suggest to modify the sentence to make it clear that different docking poses pertain to the homologs Q2 and Q4 (not to Q1).

For example, "In addition, no docking poses for KCNQ2 and KCNQ4 were similar to that seen in the ... "

Overall, it is a good manuscript and I suggest it for publication in Nature Communications.

REVIEWERS' COMMENTS

Reviewer #1 (Remarks to the Author):

The authors have done a good job in revising the manuscript and most of my previous concerns have been addressed. The authors now include extended discussions about the mechanisms underlying altered ML277 sensitivity of KCNQ1 mutations and putative mechanistic implications of the C-terminal twist and KCNE3. However, in the revised sections, I have some additional suggestions of clarifications to make:

1. I like that the authors have employed MD simulations on a version of the open KCNQ1 structure to further explore putative ML277 interaction with different conformational states and the impact of PIP2. However, some aspects of how the MD simulations were done are poorly described in the main text and SI methods. Please add necessary details such as: “its binding site refinement”, what is this?; “Optimized using ICM-Pro”, what is this; “the tetramerization domain was trimmed”, please indicate from which residue subsequent protein structure was omitted.

“its binding site refinement” and “Optimized using ICM-Pro” are now described in Results and Methods as minimization of protein-ligand complex and search for the energetically more favorable state. In MD simulations, for Q1-ML277, we used residues ASN95-GLN349 and for Q1-PIP2-ML277 residues THR104-VAL355. This has been added to the Methods.

2. Similarly, for docking, it is hard to understand from the main text and figure legends when docking was made or when structures were just superimposed. Moreover, it is not clear to me whether docking with flexible side chains was always used (e.g., also when exploring the chi angle (row 250-256). Please make clear in each figure legend when data represent resolved structures with ML277 binding, docking of ML277 to structures to predict putative binding, or just superposition of structures to indicate putative differences in binding sites.

In all cases, we used docking with flexible sidechains. For exploring the F270/F260 chi1 angle, docking was not used. A note has been added to the Methods to state this. We changed chi1 values for this residue to find conformations with steric clashes.

3. Row 244-264: The revised text on how KCNE3 may render the KCNQ1 channel insensitive to ML277 is improved. However, given the constitutively open phenotype of KCNQ1/KCNE3, it is still not clear to me whether one could expect ML277 effects on KCNQ1/KCNE3 (even if specific side chains were in a favourable rotation). Please comment.

Yue et al (2013) do show effects of ML277 increasing current amplitude in 1:1 and 1:10 KCNQ1:KCNE3 oocyte co-injections (their Fig.4), although we cannot know the exact expressed ratios. So, ML277 can affect KCNQ1:KCNE3 when it is not prevented from reaching its binding site. We have added a note in the Results to indicate that saturating amounts of KCNE3 are required to prevent the action of ML277.

4. Row 437-439: “We take some comfort that the binding pocket that we have identified does not change much from the closed... to the open structure”. Could the authors comment on how they

take comfort in this? Given the ML277 effects on KCNQ1, wouldn't it make more sense if the binding energy was greater in the open state?

There is no suggestion from electrophysiological studies that ML277 unbinds during the open-closed-open activation cycle, so we were comforted that the binding site was not much changed as it suggests the continued presence of a rational binding site for the drug. We have added this note to the Results text.

5. Row 195 and 402: The authors refer to the “native structure”. Can they please clarify this? Is this the unbound apo structure?

This has been changed to ‘apo’

6. Row 283: “ICM”, please define abbreviation.

ICM is the name of the software and we now define it as "Internal Coordinate Mechanics" in the text at this place

7. Row 327: “reflects changes in the kinetics of VSD activation”, please clarify how the deactivation kinetics mainly reflects VSD activation kinetics.

We have changed ‘activation’ to ‘movement’.

8. Row 507: “electromechanical coupling (EMC)”, this abbreviation was already stated on row 502.

Removed

9. Row 534: “activated state”, do the authors mean open state?

OK

10. It is great that the authors use a three-letter indication for xKCNQ1 residues and one-letter indication for hKCNQ1 residues. However, in the Discussion, there seems to be some mix-up (e.g., row 480 and 485). Please double-check if the correct indication is used throughout the Discussion.

Checked

11. Figure 5 legend title: “regulators to KCNQ1”. Perhaps change to KCNQ as not all regulators act on KCNQ1.

We changed this to KCNQ1 and KCNQ4 since those are the structures shown.

12. Supplementary figure 9: “Note that sidechain orientation now matches that seen in...”. Is the sidechain re-orientation caused by KCNE3 removal and subsequent relaxation of the model, or

by relaxation per se? In other words, is the same re-orientation seen when running MD in the presence of KCNE3?

Relaxation or energy minimization itself cannot cause this kind of conformational change. In Supplementary Fig.9 (now Suppl. Fig.8), we showed the results of MD simulations without KCNE3 and in the presence of ML277. Also, as we mentioned in the manuscript, the absence of KCNE3 is a necessary condition for the rearrangement of F260/F270. The conformation of F260/F270 seen in our cryoEM structure is impossible in the presence of KCNE3 because of steric clashes. In addition, we did not perform MD simulations of KCNQ1:KCNE1 since the correct binding of ML277 to KNCQ1 is impossible when the sidechain of F260/F270 is oriented toward the binding site.

Reviewer #2 (Remarks to the Author):

The manuscript has been greatly improved and the revision was met with addition of significant amounts of new data. The updated Figures 2, 3, 4 are more illustrative. In this revised manuscript, the authors address previous concerns from me and other reviewers related to the binding pose of the ML 277 in the open channel, when the PIP2 molecule is also bound, as well as to the corresponding mechanism of action. These issues have been satisfactorily addressed by docking experiments and molecular dynamics simulations on open channel (PDF ID 6v01). The “ML 277 mechanism of action” section in the Discussion now reasonably explains the issues I raised related to ML277 action rendering the channel constitutively open. Discrepancies of drug action on different mutants, including V255A and S338A, are discussed in the Discussion. I found only one error described below.

Page 10, line 223: Xua et al. 2014. The publication year should be changed to 2015.

Corrected

Reviewer #4 (Remarks to the Author):

The authors have appropriately responded to my questions. No further revision is needed from me.

Reviewer #5 (Remarks to the Author):

It is a great article with a discovery of the binding model of this potassium channel activator ML277.

Some comments on the binding energy calculations using the obtained structural model and MMPBSA.

I understand that the numbers are beyond the scope of this manuscript but binding free energies around -50 kcal/mole are clearly grossly exaggerated for a small molecule binder such as ML277 since it translates into insanely strong binding constant (Kd orders of magnitude below

femtomolar). I am not sure anything can be done about it however by the authors.

We agree with this comment, -50 kcal/mol is a very strong binding energy, but this is what we get using MMPBSA with an implicit membrane. The same calculations without the implicit membrane, (just implicit water), give a binding energy around -30 kcal/mol. We preferred to use an implicit membrane since the binding site is in the transmembrane region.

The docking of ML277 and some other ligands to the KCNQ1 model and its homologs (Q2,Q4) are very useful to understand the specificity and alternative binding pockets. It was great to see that ICM docking predicted binding mode close to the experimentally determined one.

I suggest some language changes:

l. 183: "which strongly supports the posited location for ML277" , I assume it is a typo, may be the authors meant "proposed"

Changed

line 283,284 : I suggest to say, "less favorable" instead of "lower". Term "lower" implies more favorable binding and -31 lower than -16.6, not higher since these are negative values. Using "less favorable" is one way to avoid going into all those details yet keep the meaning clear. The ICM docking scores for KCNQ2 and KCNQ4 were -16.6 and -9.42, respectively, which is much less favorable than for xKCNQ1-ML277 (-31.1).

Changed

line 285. I suggest to modify the sentence to make it clear that different docking poses pertain to the homologs Q2 and Q4 (not to Q1).

For example, "In addition, no docking poses for KCNQ2 and KCNQ4 were similar to that seen in the ... "

Text changed to that suggested.

Overall, it is a good manuscript and I suggest it for publication in Nature Communications.